# Concerted localization-resets precede YAP-dependent transcription

J. Matthew Franklin[1,2,3,4,5,6], Rajarshi P. Ghosh [1,2,3,4,6 ✉], Quanming Shi[1,2,3,4], Michael P. Reddick[1,2,3,4,5] & Jan T. Liphardt [1,2,3,4 ✉]

Yes-associated protein 1 (YAP) is a transcriptional regulator with critical roles in mechanotransduction, organ size control, and regeneration. Here, using advanced tools for real-time visualization of native YAP and target gene transcription dynamics, we show that a cycle of fast exodus of nuclear YAP to the cytoplasm followed by fast reentry to the nucleus ("localization-resets") activates YAP target genes. These "resets" are induced by calcium signaling, modulation of actomyosin contractility, or mitosis. Using nascent-transcription reporter knock-ins of YAP target genes, we show a strict association between these resets and downstream transcription. Oncogenically-transformed cell lines lack localization-resets and instead show dramatically elevated rates of nucleocytoplasmic shuttling of YAP, suggesting an escape from compartmentalization-based control. The single-cell localization and transcription traces suggest that YAP activity is not a simple linear function of nuclear enrichment and point to a model of transcriptional activation based on nucleocytoplasmic exchange properties of YAP.

[1] Bioengineering, Stanford University, Stanford, CA 94305, USA. [2] BioX Institute, Stanford University, Stanford, CA 94305, USA. [3] ChEM-H, Stanford University, Stanford, CA 94305, USA. [4] Cell Biology Division, Stanford Cancer Institute, Stanford, CA 94305, USA. [5] Chemical Engineering, Stanford University, Stanford, CA 94305, USA. [6] These authors contributed equally: J. Matthew Franklin, Rajarshi P. Ghosh. ✉email: ghosh.rpg@gmail.com; jan.liphardt@stanford.edu

Transcriptional effectors that shuttle between cellular compartments often encode signaling cues in their shuttling dynamics[1]. A prime example of a signal transducer that shows differential compartmentalization in response to various physiological cues is the YAP (YES-associated protein)/TAZ (transcriptional coactivator with PDZ-binding motif) duo[2]. YAP/TAZ constitutes a central node of the Hippo pathway[3,4], controls organ size, and is critical for mechanotransduction[5–7]. The classic view of YAP signal transduction equates nuclear enrichment with activation of pro-growth transcriptional programs through association with the TEAD family of transcription factors (TFs)[8–10]. However, other critical signal transducers such as ERK, NfkB, and P53 use a rich signal transmission code in which the amplitude, frequency, and duration of their nucleocytoplasmic shuttling all influence gene transcription[11–13]. Dysregulated YAP signaling has been implicated in several forms of cancer, although conflicting views exist[8], pointing to a gap in mechanistic understanding.

A recent study of Yorkie (Drosophila YAP homolog) subcellular localization and dynamics in *D. melanogaster*[14] and several studies using live imaging in mammalian cells expressing exogenous YAP[15,16] have reported rapid shuttling between cytoplasm and nucleus. Further, using immunostaining, Lin et al. showed that TEAD translocated rapidly to cytoplasm under stress in Hek293 cells[17]. Given these reports, we speculated that YAP/TAZ and TEAD subcellular spatiotemporal dynamics might encode upstream signaling information into transcriptional outputs. More generally, we should not expect biological signal transmission circuits to represent simple a linear mapping of an external input onto the downstream gene expression level. There are now many examples of gene circuits with hysteresis, memory, and latching[18–22]. Since these circuits involve spatially controlled components such as TFs, it is natural to wonder about the role of spatial dynamics of YAP in cellular signal processing particularly in a disease context, such as upon Ras transformation which upregulates YAP-dependent transcription in many biological contexts[23–25].

A dynamical analysis of the relationship between the regulator module and the responder module of a signaling pathway would require real-time tracking of both TF localization dynamics and nascent transcription dynamics of endogenous alleles. To quantify the localization dynamics of native YAP/TEAD in single cells and relate such dynamics to downstream transcription, we used CRISPR to fluorescently tag native *YAP* and *TEAD* genes (regulators), as well as the mRNA of two well-documented YAP target genes, *ANKRD1* and *AREG* (responders) with a 24X MS2 transcriptional reporter cassette.

In the current work using live-cell imaging and quantification of protein localization and transcriptional activity, we delineate regulatory mechanisms that govern the relationship between YAP localization and YAP-dependent transcription. First, we show that nontransformed epithelial cells, as well as human embryonic stem cells, exhibit spontaneous fluctuations in YAP localization. A variety of input-parameters modulate these fluctuations, including HRas transformation, calcium signaling, actomyosin contractility, mitotic exit, and mechanical defects in the nuclear lamina. We also show "localization-resets", which are single cycles of rapid YAP exodus from the nucleus to cytoplasm followed by fast reentry back into the nucleus on time scale of 0.5–2 h. In nontransformed epithelial cells, transient YAP localization-resets correlate highly with transcriptional activation of YAP target genes. Ras-transformation markedly dampens YAP fluctuations. Using nucleocytoplasmic transport analysis, we show that HRas transformation dramatically increases nucleocytoplasmic turnover of YAP while reducing bulk-chromatin interactions as determined by spatiotemporal FRAP (fluorescence recovery after photo-bleaching). Patient derived triple negative breast cancer lines harboring either HRas or KRas mutations also show this defect. Together, these results suggest a model of transcriptional activation gated by tight control of YAP localization, where Ras transformation bypasses this control allowing permanent activation of the YAP transcriptional program.

## Results

**Genome knockin lines reveal endogenous YAP dynamics.** The current model of YAP transcriptional control proposes a linear relationship between nuclear YAP localization and downstream gene activation. However, this is largely based on bulk biochemical assays and immunostaining of fixed cells. Our initial goal was (i) to track the dynamics of native YAP localization across a broad range of timescales in response to different signaling cues and (ii) to assess how alterations in YAP localization are imprinted onto the transcriptional control of YAP target genes. Using CRISPR-Cas9[26,27] (see "Methods", Supplementary Table 1), we generated a *YAP*-eGFP knock-in cell line from the breast epithelial cell line MCF10A (MCF10A$^{YAP\text{-}GFP\text{-}KI}$, Fig. 1a). Genomic PCR showed proper insertion of an eGFP-p2a-puromycin cassette at the C-terminus of YAP (Supplementary Fig. 1b, see "Methods", Supplementary Table 2), and revealed amplicons of two different sizes corresponding to knockin and wildtype alleles suggesting that the base MCF10A$^{YAP\text{-}GFP\text{-}KI}$ cell line is a mixture of heterozygous and homozygous knockins. Capillary western blot (see "Methods" section) against eGFP showed a single predominant band of expected molecular weight, indicating specific tagging of YAP and efficient cleavage of the puromycin from the YAP-eGFP fusion by the self-cleavable P2A peptide sequence (Fig. 1c). Following puromycin selection, we used immunostaining of YAP to confirm the accurate localization of the YAP-eGFP fusion. We found the nucleocytoplasmic ratio (N/C) of eGFP and native-YAP staining (Fig. 1c, d) were highly correlated on a per-cell basis, indicating proper integration of the eGFP tag.

To ensure that the eGFP fusion did not perturb the localization of YAP, we performed a series of tests based on prior work that has shown changes in YAP localization in response to specific signaling cues. YAP is a key mechano-transducer that translocates from the cytoplasm to the nucleus in response to an increase in extracellular matrix substrate stiffness[9]. Indeed, when seeded on polyacrylamide gels with lower stiffness (0.4 kPa), YAP showed strong cytoplasmic enrichment, whereas, at higher stiffness (60 kPa), YAP showed enhanced nuclear distribution (Fig. 1e). Quantification of YAP N/C over a range of substrate stiffnesses showed statistically significant differences in YAP distribution. To test the ability of YAP to switch compartments in response to well characterized perturbations, we tracked YAP localization during the disruption of the actin cytoskeleton[9]. Indeed, inhibition of F-actin polymerization by Latrunculin B led to cytoplasmic sequestration of YAP as previously reported (Fig. 1g, h and Supplementary Movie 1)[28]. While comparison to known modulations of YAP localization demonstrates the functional veracity of the YAP-eGFP knock-in, subtle effects of tagging should be tested for specific biological context.

A recent report has shown that TEAD, the TF partner of YAP, undergoes cytoplasmic sequestration in HEK 293a cells at high cell densities[17]. To simultaneously track YAP and TEAD subcellular localization in real-time, we generated a dual CRISPR knock-in MCF10A cell line, where native *YAP* and *TEAD1* (the most abundant TEAD family member in MCF10A)[29] were genomically tagged with eGFP and mCherry, respectively. Genomic PCR showed proper insertion of a mCherry-p2a-hygromycin cassette at the C-terminus of TEAD (Supplementary

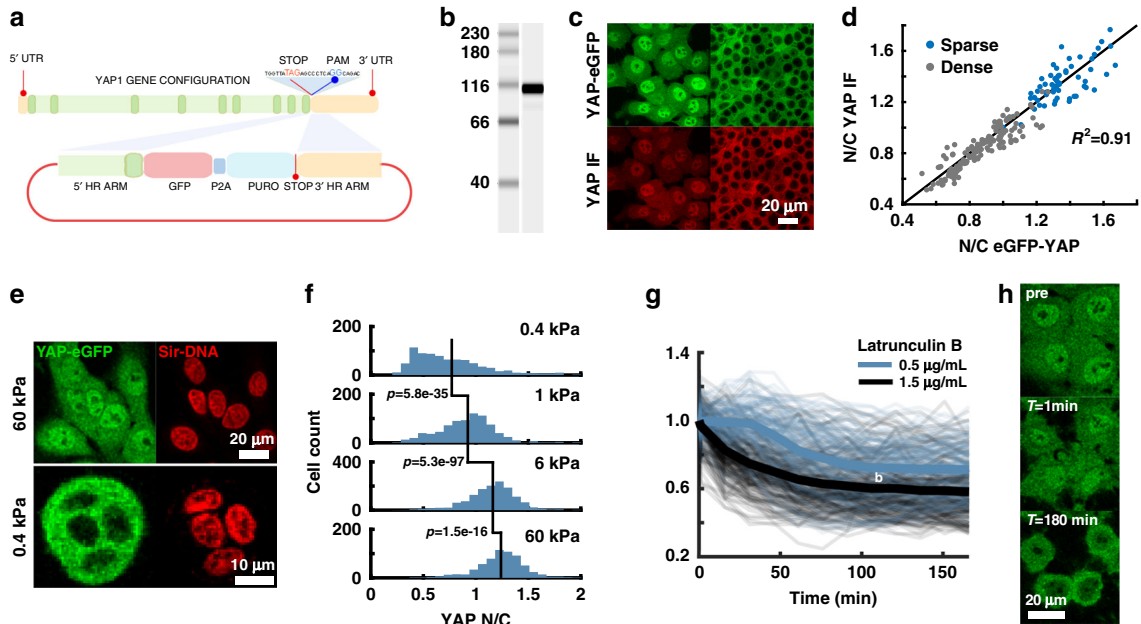

**Fig. 1 Characterization of YAP and TEAD genome-knockin cell lines. a** Cartoon of CRISPR-Cas9 based insertion of eGFP-P2A-puromycin cassette at 3′ end of the YAP gene. **b** western blot against eGFP from MCF10A$^{YAP\text{-}GFP\text{-}KI}$ whole cell lysate. Single predominant band at the predicted YAP-eGFP fusion size 96 kDa. Repeated twice with similar results. **c** Correlation of the YAP nuclear/cytoplasmic ratio ($N/C$) signal for YAP-eGFP and anti-YAP immunofluorescence in sparse and dense cultures. $N_{sparse} = 61$ cells, $N_{dense} = 155$ cells from one experiment. Repeated twice with similar results. **d** Representative Immunofluorescence images of data from **c. e** MCF10A$^{YAP\text{-}GFP\text{-}KI}$ cells plated on soft (0.4 kPa) and stiff (60 kPa) acrylamide showing altered cell morphology and YAP localization. Cells were counterstained with the live-cell nuclear stain, SiR-DNA™. **f** Quantitative comparison of YAP-eGFP $N/C$ on fibronectin coated acrylamide of varying stiffnesses. $N_{0.4\ kPa} = 834$, $N_{1\ kPa} = 818$, $N_{6\ kPa} = 1386$, $N_{60\ kPa} = 647$ cells. A one-sided Mann–Whitney $U$-test was used to test differences in YAP $N/C$. 0.4 kPa vs. 1 kPa: $5.8 \times 10^{-35}$, Ranksum = 570,253. 1 kPa vs. 6 kPa: $p = 5.3 \times 10^{-97}$, Ranksum = 600,652. 6 kPa vs. 60 kPa: $P = 1.452 \times 10^{-16}$, Ranksum = 1,308,747. Data compiled from two independent experiments for each condition. **g** Time-course of YAP $N/C$ normalized to initial value following Latrunculin B treatment at 0.5 μg/mL (140 cell traces) and 1.5 μg/mL (256 cell traces). Individual cell traces in transparent lines with mean plotted as solid line. Data compiled from one independent experiment for each condition. **h** Fluorescent images of YAP-eGFP before and after Latrunculin B treatment for the indicated times.

Fig. 1b and Supplementary Table 2). While YAP showed significant cytoplasmic sequestration at high densities, TEAD1 localization remained nuclear (Fig. 2a, b). In the remainder of the study, we therefore focused on YAP as the dominant nucleocytoplasmic signal transducer in the Hippo pathway.

Endogenous tagging provided the unique opportunity to dissect YAP localization dynamics without confounding effects of exogenous expression, which decouples TF expression from the cell's intrinsic regulatory machinery. Simultaneous tracking of local cell density and YAP localization revealed that sparsely plated monolayers of MCF10A$^{YAP\text{-}GFP\text{-}KI}$ maintain relatively constant YAP $N/C$ up to a local density threshold, after which $N/C$ decreases with a concomitant increase in local cell density (Fig. 2c and Supplementary Movie 2). At low density, newly divided cells maintained constant local cell density by migrating to void-spaces. Further division cycles depleted the voids, packing the cells and driving cytoplasmic sequestration of YAP.

Since HRas transformation inhibits the *hippo* pathway[30], we asked whether HRas-transformation altered YAP localization in MCF10A cells[31]. In line with previous findings, we found that HRas-transformation abolished YAP cytoplasmic sequestration at high density with a smaller change in YAP $N/C$ as cells grew from sparse to confluent (Fig. 2d and Supplementary Movie 2)[30]. Interestingly, we found that HRas transformation alters the packing density of MCF10A, in addition to its known effect on growth rate (Supplementary Movie 2). To account for these changes, we plotted the YAP $N/C$ measurements over the same range of local cell densities for MCF10A vs. HRas (Fig. 2c, e). This clearly shows that at equivalent high density, HRas transformed

cells have impaired nuclear exclusion. HRas transformation has been shown to decrease LATS1/2 activity[30], which is the primary kinase involved in density sensing and cytoplasmic sequestration of YAP[7,8]. The decreased dynamic range of YAP $N/C$ after HRas transformation indicates that the Hippo pathway is impaired.

To test if eGFP tagging retained functional competence of endogenous YAP in cells other than immortalized nontransformed cell lines, we generated the YAP-eGFP knockin of H1 human embryonic stem cells (hESC$^{YAP\text{-}GFP\text{-}KI}$). YAP phosphorylation levels and cytoplasmic localization have been shown to increase at high cell densities in human pluripotent stem cells (H9 hESCs and 19-9-11 iPSCs)[32]. Lower expression levels of native YAP in H1 hESCs prohibited prolonged (multiday) live-cell imaging due to cumulative photo-toxicity to the cells at higher laser powers necessary for reliable tracking of $N/C$. However, when imaged four days after seeding on matrigel, a confluent culture of hESC$^{YAP\text{-}GFP\text{-}KI}$ (seeding density of $0.5 \times 10^5/cm^2$) showed distinct nuclear enrichment of YAP, whereas cells seeded at an eight fold higher density (seeding density of $4 \times 10^5/cm^2$) showed a marked decrease YAP $N/C$ (Fig. 2f, g).

**Various signaling cues modulate YAP spatial-fluctuations**. A close inspection of individual cells during monolayer growth revealed large fluctuations in YAP $N/C$, with localization inverting within 120 min (Fig. 3a and Supplementary Movie 3). To characterize these localization fluctuations, we measured YAP $N/C$ ratios over a 24-h period at a 15-min sampling frequency. Single cell localization traces revealed rapid changes in YAP $N/C$

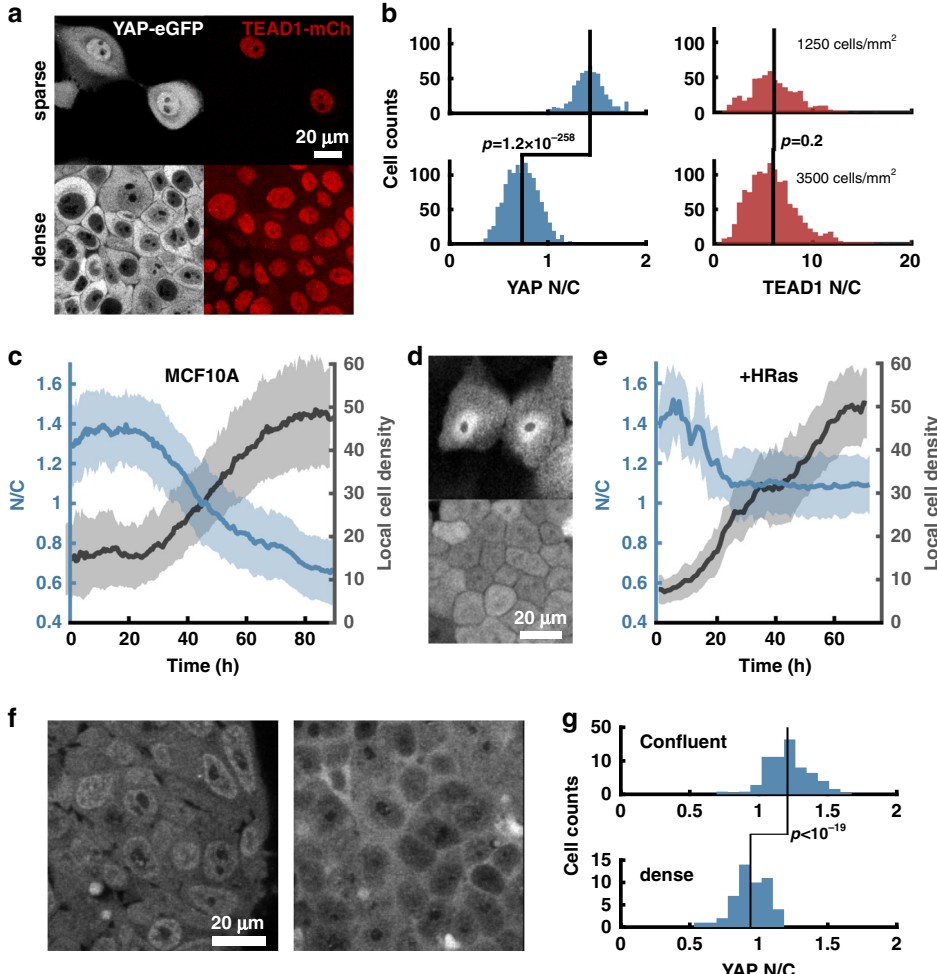

**Fig. 2 YAP and TEAD localization dynamics: Density sensing. a** Dense and sparse MCF10A cells with YAP-eGFP and TEAD1-mCherry dual CRISPR knockin. **b** N/C of YAP and TEAD1 in MCF10A at various densities. $N = 584$ for sparse condition (1250 cells/mm²), and $N = 1272$ for 3500 cells/mm². A one-sided Mann–Whitney U-test was used to test differences in N/C ratio between conditions. For YAP sparse vs. dense: $P = 1.2e-258$, Zval: 34.3, Ranksum = 910,364. For TEAD sparse vs. dense: $P = 0.23$, Zval = 1.1930, ranksum = 555,036. Repeated three times with similar results. **c** Time-lapse of YAP-eGFP nuclear/cytoplasmic ratio (N/C) during monolayer growth for MCF10A (solid line is mean value, and shading is standard deviation) ($N = 915$ cell tracks). Data from one representative experiment. Repeated three times with similar results **d** Native YAP-eGFP expression/localization in sparse and dense cultures of MCF10A-HRas (MCF10AT), highlighting lack of YAP nuclear exclusion at high density. **e** Time-lapse of YAP-eGFP nuclear/cytoplasmic ratio (N/C) during monolayer growth for MCF10AT ($N = 1478$ cell tracks, solid line is mean value, and shading is standard deviation). Data from one representative experiment. Repeated three times with similar results. Neighbors in **c** and **e** is the average local cell density, defined as the number of cell centroids located with a 250-pixel radius. **f** Representative images of H1 hESC$^{YAP\text{-}GFP\text{-}KI}$ cells at confluent and very high density. **g** Quantification of YAP N/C in experiments from **f**. A one-sided Mann–Whitney U-test was used to test differences in YAP N/C. $N_{confluent} = 161$, $N_{dense} = 50$ cells. $P = 2.66 \times 10^{-7}$, Ranksum = 18,761. Repeated twice with similar results.

ratio (time scale of 30–60 min), suggesting that YAP localization is dynamically tunable (Fig. 3b, c).

As YAP-localization is primarily controlled through phosphorylation[33], we hypothesized that modulating upstream kinase activity would alter YAP N/C dynamics. We tested this possibility via HRas$^{G12V}$ transformation which suppresses LATS activity, and Src kinase inhibition which upregulates LATS activity[30,34,35]. We found that HRas transformation appeared to dampen YAP localization dynamics (Fig. 3c and Supplementary Movie 4). Previous reports have shown uniform and permanent changes in YAP localization after Src inhibition in serum-starved cells[34], and when used in conjunction with Trypsinization-reattachment cycle[35]. To decouple the effect of Src kinase inhibition from other forms of modulations such as serum starvation[36] or cell detachment/adhesion cycles[37], we treated subconfluent MCF10A cells grown in complete media with PP1, a potent inhibitor of Src kinase[38]. Surprisingly, we found that treatment with 1 μM PP1 in complete growth media lead to exaggerated localization fluctuations rather than uniform cytoplasmic sequestration (Fig. 3c and Supplementary Movie 4). Tracking of YAP N/C over time allowed us to quantify these fluctuations by detecting rapid changes in YAP N/C and assess if these changes were statistically relevant. The fraction of cells with at least one fluctuation increased upon PP1 treatment (56% of cells) and decreased upon HRas transformation (6% of cells) compared to untreated MCF10A cells (23% of cells, Fig. 3d). Further, we characterized the frequency (number of detected fluctuations per time) and amplitude (net change in N/C) of these fluctuations. The fluctuation frequency and amplitude were unchanged upon HRas transformation (Supplementary Fig. 2a, b). On the other hand, PP1 treatment led to a slight, but significant, increase in both fluctuation frequency and amplitude (Supplementary Fig. 2a, b).

To measure YAP localization dynamics under conditions where a single parameter switch in ambient conditions causes a

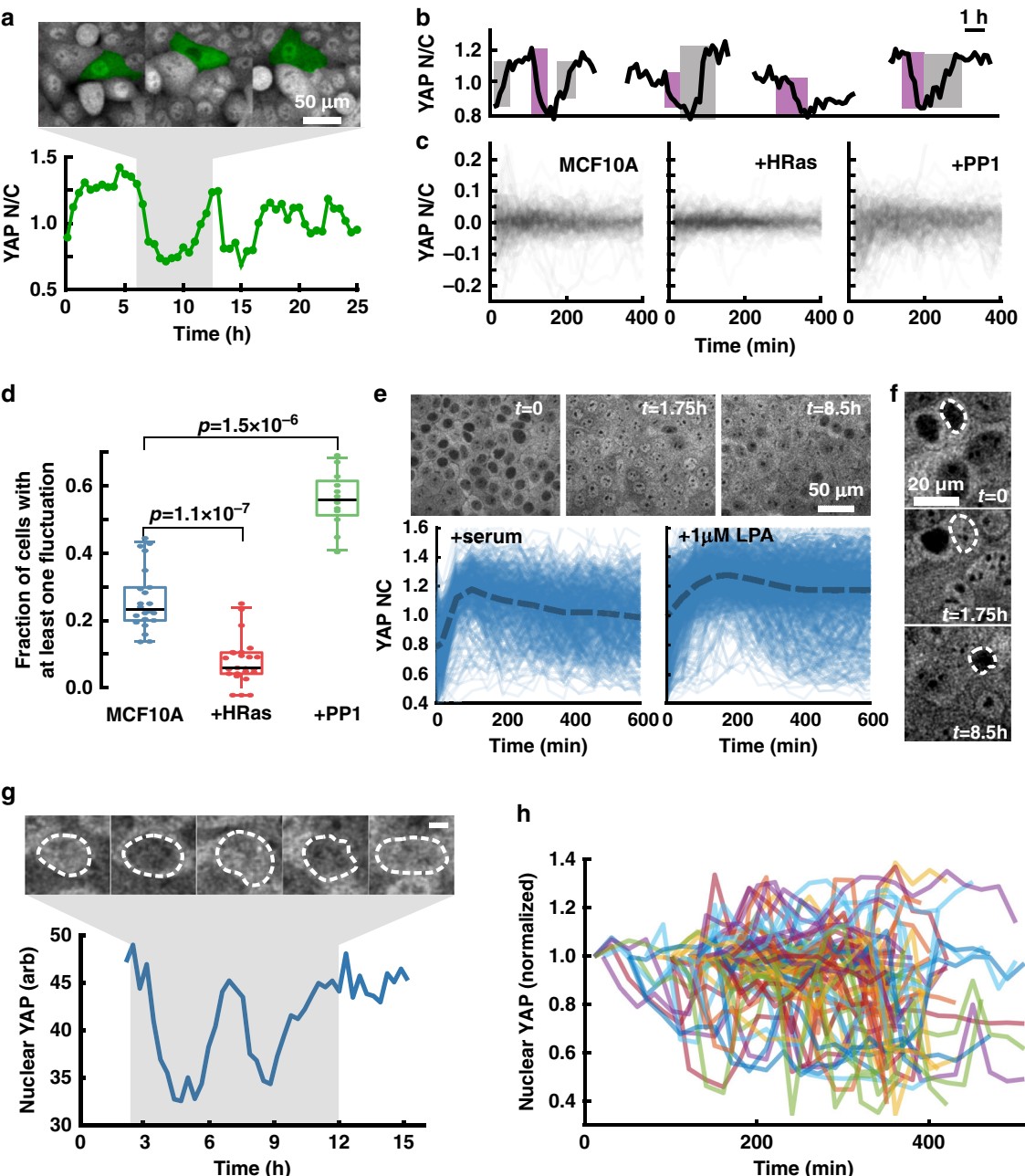

**Fig. 3 Baseline (minute-scale) fluctuations in YAP localization. a** *N/C* trace of an individual MCF10A cell in monolayer and corresponding fluorescence image (cell of interest is false colored in green). **b** Example traces of YAP *N/C* over time with large changes in signal highlighted by gray (increase) and purple (decrease) in MCF10A cells. **c** Overlay of YAP *N/C* traces for MCF10A, MCF10A + HRas, and MCF10A + 1 μM PP1. Each plot contains 200 traces randomly selected from the pool of all tracks collected from each condition. The mean *N/C* value is subtracted from each trace to allow visual comparison. **d** Mean fraction of cells with at least one detectable fluctuation within a 24 h time-course (24 ROIs for each condition compiled from two independent experiments). Box and whisker lines: black-center line is median value, box-bounds are 25th and 75th percentile, and whiskers are 0.05 and 0.95 percentiles. A one-sided Mann–Whitney *U*-test was used to test whether the fraction of cells displaying a significant fluctuation was changed after HRas expression ($N = 24$, $P = 9.0 \times 10^{-7}$, Ranksum-Stat = 734) or PP1-treatment ($N = 24$, $P = 3.52 \times 10^{-4}$, Ranksum-Stat = 290.5). **e** (Top panel) Time-course of MCF10A$^{YAP\text{-GFP-KI}}$ cells following 20 h serum starvation followed by serum replenishment. (Bottom panel) Quantification of YAP N/C following stimulation by serum or 1 μM LPA stimulation. **f** Fluorescent images of serum starved cells followed by serum stimulation for the indicated times. **g** Fluctuation of nuclear YAP in a single representative H1 hESC$^{YAP\text{-GFP-KI}}$ cell. (Top panel) Nuclear outline highlighted in dashed line. (bottom panel) Quantification of nuclear YAP intensity over time. At very high density in hESC$^{YAP\text{-GFP-KI}}$ cells it is not feasible to reliably estimate cytoplasmic intensity. **h** Overlay of nuclear YAP intensity in many individual hESC$^{YAP\text{-GFP-KI}}$ cells showing dynamic changes in intensity levels. Repeated twice with similar results.

dramatic redistribution of YAP, we imaged YAP in serum-starved cultures before and after serum replenishment or treatment with the GPCR agonist lysophosphatidic acid (LPA)[36]. Utilizing our ability to capture and quantify YAP localization in real-time, we asked whether cells respond uniformly, or are there subtler dynamics at the level of individual cells. As expected, serum-starved cells were initially devoid of YAP, and both serum replenishment and LPA treatment induced rapid nuclear accumulation of YAP (Fig. 3e). Interestingly, the response decorrelated over time, with many cells exhibiting nuclear

depletion of YAP (Fig. 3e and Supplementary Movie 5). By following individual cells, we found many examples of rapidly changing YAP localization, reminiscent of the spontaneous fluctuations of YAP in unperturbed monolayer cultures (Fig. 3f). These results demonstrate the heterogeneous response landscape of individual cells.

Next, we investigated whether these dynamic fluctuations in YAP localization occurred in H1 hESCs. Indeed, we found frequent oscillations in YAP localization at time scales similar to MCF10A (Fig. 3g, h and Supplementary Movie 6), demonstrating that these fluctuations occur in primary cell culture.

**Intracellular calcium release induces rapid YAP localization-reset.** Fluctuations in biological systems are often considered crucial for functional and phenotypic "plasticity"[39]. To delineate the temporal scales over which YAP localization dynamics encode large and rapid changes in environment, we investigated how YAP-localization changes upon monolayer wounding. Previous immunofluorescence studies demonstrated fast (~30 min) nuclear translocation of YAP at the wound edge of mammary epithelial cells[7]. Surprisingly, real-time tracking of YAP immediately after wounding the monolayer revealed an oscillatory response: rapid nuclear accumulation in edge cells (~1 min), followed by depletion (lasting ~20 min), and finally slow nuclear reaccumulation (~3 h, Fig. 4a and Supplementary Movie 7).

The rapid fluctuations in YAP localization upon wounding hinted at a fast-acting upstream signaling pathway. Both mechanical deformation of cells at the wound edge and a dramatic shift in local cell density could potentially contribute to the oscillation in YAP localization. A fast calcium wave (FCW) within seconds of epithelial wounding has been reported in diverse cell types, including MCF10A[40–42]. We hypothesized that intracellular calcium may play a role in the rapid nuclear accumulation of YAP upon wounding. To test this possibility, we released intracellular calcium via Thapsigargin (TG)[43] treatment. This store operated calcium entry (SOCE) resulted in a bulk YAP translocation cycle: an initial fast depletion (lasting ~25 min) followed by slow nuclear enrichment (50 min, Fig. 4b and Supplementary Movie 8). Although the wound assay showed a distinct pattern with an initial phase of nuclear translocation, TG treatment helped us isolate the effects of transient $Ca^{2+}$ release on YAP from the multitude of parallel signals occurring during the wound response. The $Ca^{2+}$ dynamics after TG treatment were tracked using the fast kinetic $Ca^{2+}$ sensor GcAMP6f[44], revealing a rapid increase in intracellular $Ca^{2+}$ matching the onset time of nuclear YAP depletion, followed by a slow decay, and then a sustained low amplitude oscillatory phase (Fig. 4c). Such collective $Ca^{2+}$ oscillations have been demonstrated at monolayer wound edges[45].

Ionomycin, a $Ca^{2+}$ ionophore that also increases intracellular $Ca^{2+}$, showed similar effects to TG (Supplementary Fig. 3a). Simultaneous treatment of ionomycin and the potent phosphatase inhibitor Okadaic acid, which inhibits protein phosphatase 1 upstream of YAP[46], led to a similar degree of cytoplasmic sequestration but with minimal recovery (Supplementary Fig. 3a). This result hints at the possibility that $Ca^{2+}$ may drive the phosphorylation of YAP, followed by PP1-dependent dephosphorylation. Previous reports have shown that the release of extracellular ATP may drive FCW at epithelial wound edge[47]. Although addition of 10 mM extracellular ATP induced a similar response as TG and ionomycin, we found there was a sharp dependence on cellular density, with sparse cells responding minimally (Supplementary Fig. 3b). A recent paper demonstrated that Protein kinase C (PKC) beta II mediates $Ca^{2+}$-driven phosphorylation and cytoplasmic sequestration of YAP[48] in

glioblastoma cells, through activation of Lats 1/2 kinases. While the role of PKCs in YAP regulation has been previously explored, the nature of the regulation seems to be context and cell type specific[49]. For example, TPA induced activation of PKCs have been shown to dephosphorylate YAP causing nuclear accumulation in Hek293a, HeLa and U251MG cells, but have the exact opposite effect on Swiss3T3 cells, MEF cells and the lung cancer A549 cells[49]. To test the role of the PKC-LATS axis in mediating calcium-driven localization reset, we treated MCF10A$^{YAP}$-GFP-KI cells with the potent protein kinase C inhibitor Go6976[48,49]. This caused a distinct delay in the onset of YAP nuclear depletion and reduction in the magnitude of the localization-reset response (Supplementary Fig. 3c and Supplementary Movie 8).

To test if eGFP tagging altered YAP localization dynamics in response to TG induced SOCE, we carried out immunostaining analysis of YAP distribution at different time points after TG treatment in MCF10A cell line. These measurements of native YAP localization-reset in parental MCF10A cell-line were consistent with the MCF10A$^{YAP}$-GFP-KI cell line (Fig. 4e). Because we suspected SOCE transiently activated LATS kinases to drive cytoplasmic translocation of YAP, we performed an immunofluorescence based analysis of YAP localization time-course during TG treatment in LATS1/2 knockout MCF10A cells[50]. As we hypothesized, LATS1/2 depletion completely abolished YAP localization reset (Fig. 4e, f). These data support the role of LATS1/2 as the primary regulator of SOCE induced YAP localization reset.

We found that the onset of $Ca^{2+}$ release correlated with a rapid decrease in nuclear volume followed by pulses of compression and relaxation corresponding to calcium oscillations (Fig. 4d). Since the nuclear membrane mechanically regulates YAP translocation[51], we hypothesized that the $Ca^{2+}$-induced compression of the nucleus may be contributing to the YAP localization-reset. To reduce the compressibility of the nucleus, we overexpressed the Δ50LaminA variant which confers increased rigidity to the nucleus[51]. Indeed, Δ50LaminA overexpression reduced the mean nuclear compression by 48% after TG treatment (Supplementary Fig. 3d, e and Supplementary Movie 9) and significantly reduced the $Ca^{2+}$-induced YAP localization-reset (Supplementary Fig. 3f), while not affecting SOCE amplitude or kinetics upon TG treatment (Supplementary Fig. 3g, h). This suggests that the mechanical regulation of the nuclear deformability contributes to YAP localization fluctuations. While our data clearly show that SOCE induces YAP localization reset through the LATS kinase regulatory axis, it is imperative to investigate the effect of transient physiological changes in calcium levels on YAP localization in various cellular contexts, especially how such changes may affect YAP target genes in the long term.

**YAP localization-resets and target gene activation show strong correlation.** Next, we asked whether YAP localization-resets (nuclear depletion followed by re-enrichment) affected the expression of YAP target genes. Indeed, YAP localization changes upon Angiotensin II stimulation have been associated with increased YAP dependent transcription[52]. Since transcription is pulsatile[39,53], relating localization dynamics to target gene activity requires real-time measurement of gene transcription. We therefore used CRISPR to knock-in a 24×-MS2 transcriptional reporter cassette[54] at the 3′ UTR of two well-documented YAP responsive genes, Ankyrin Repeat Domain 1 (*ANKRD1*, MCF10A$^{ANKRD1}$-MS2-KI)[55] and Amphiregulin (*AREG*, MCF10A$^{AREG}$-MS2-KI)[56] (Fig. 5a). This experimental design would allow us to (a) avoid the inherent delays resulting from mRNA export, translation, and fluorescent protein (FP)

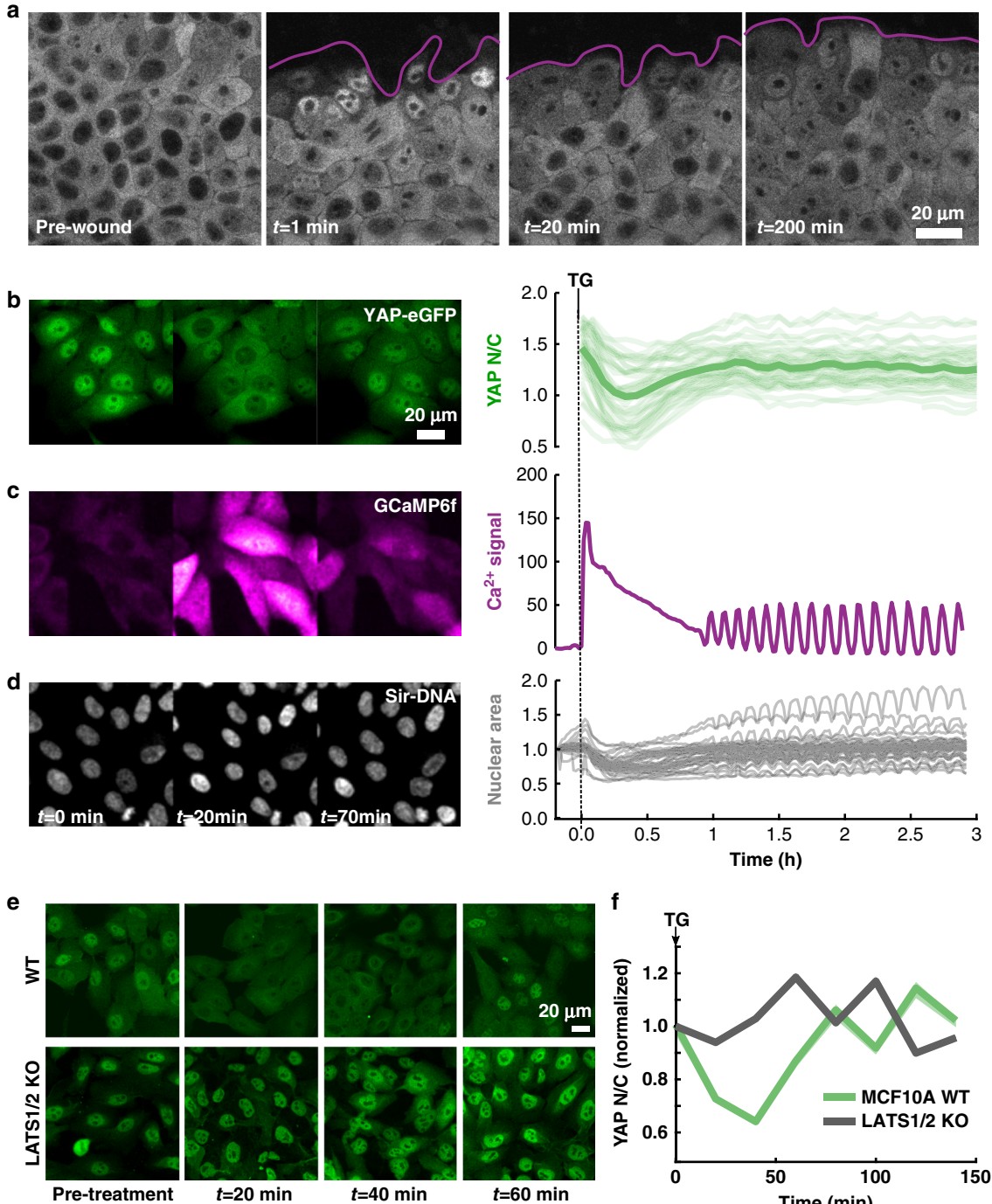

**Fig. 4 YAP translocation dynamics during wound response and calcium stimulation. a** Fluorescent images of YAP-eGFP before and after mechanical wounding. Wound edge highlighted in magenta. Data from one representative experiment. Repeated three times with similar results. **b** YAP localization, **c** cytoplasmic calcium signal, and **d** nuclear shape at different time points after induction of SOCE using 1 μM Thapsigargin. The time stamp for **b** and **c** are the same as **d**. YAP N/C traces are from 43 cells compiled from two independent experiments with similar results. Nuclear area traces and GCaMP6f Ca²⁺ traces are from 42 cells and two independent experiments with similar results. Mean value is shown with bold line; individual traces are displayed as transparent. Both calcium and nuclear area are normalized to the pretreatment signal value on a per-cell basis. **e** Immunofluorescence of native YAP in MCF10A wildtype cells and LATS1/2 double knockout cells after TG treatment for the indicated times. **f** Quantification of YAP N/C for the data in **e**. N/C ratio normalized to pre-treatment value, mean plotted in bold with shaded area as SEM. WT: $N_{pre-treatment} = 214$, $N_{20\,min} = 199$, $N_{40\,min} = 144$, $N_{60\,min} = 233$, $N_{80\,min} = 155$, $N_{100\,min} = 298$, $N_{120\,min} = 181$, $N_{140\,min} = 468$. LATS1/2 KO: $N_{pre-treatment} = 202$, $N_{20\,min} = 344$, $N_{40\,min} = 329$, $N_{60\,min} = 255$, $N_{80\,min} = 220$, $N_{100\,min} = 116$, $N_{120\,min} = 607$, $N_{140min} = 435$ cells. Data compiled from two independent experiments.

maturation times seen with FP reporters driven by YAP-responsive promoter-arrays[9], (b) avoid potential artifacts due to position effects[57] that can occur through random incorporation of synthetic reporters into the genome, and (c) provide a realistic biological measurement by tracking native genes that are regulated by YAP/TEAD.

We first verified the proper insertion of the MS2 cassette using genomic PCR (Supplementary Fig. 4 and Supplementary Table 2).

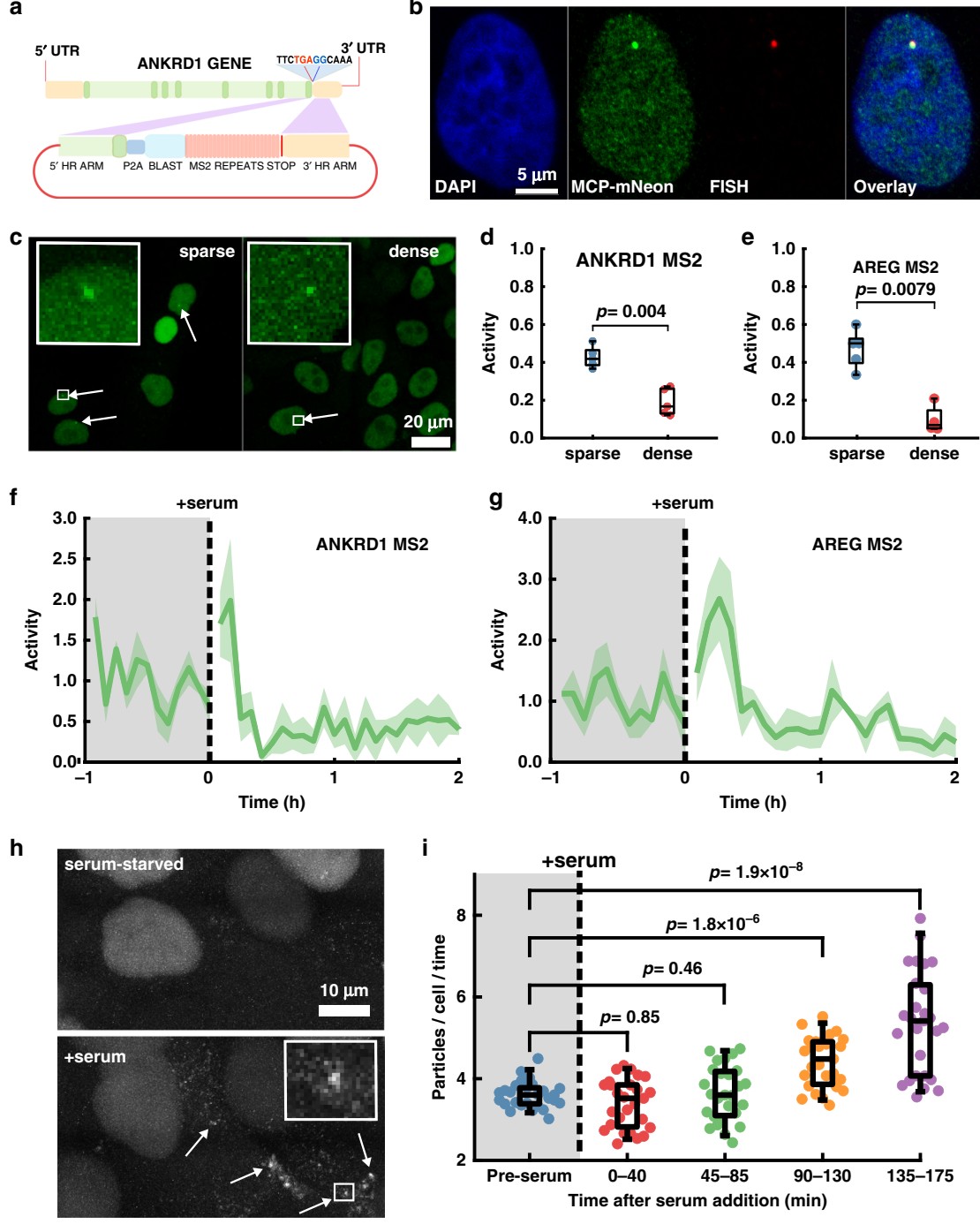

Co-expression of a mNeon-Green fusion of the bacteriophage MS2 coat protein (MCP-mNeon-NLS)[54] allowed us to quantify instantaneous transcriptional output by monitoring transcribing loci in real time (Fig. 5b). To test the veracity of our nascent transcript detection platform, we carried out single-molecule RNA FISH against native AREG transcripts in MCF10A^AREG-MS2-KI coexpressing MCP-mNeon. We found a strict colocalization of the MCP-mNeon signals and Quasar 670-labeled oligonucleotides, indicating our read-out is specific to our gene of interest (Fig. 5b). For ANKRD1, overwhelming background signal, most likely due to nonspecific oligonucleotide binding precluded detection of transcripts at single-molecule resolution.

To further verify if the nascent transcription read-out from MCF10A^AREG-MS2-KI recapitulated results from earlier studies on

transcriptional regulation by YAP, we compared nascent transcription in sparse and dense cultures of these reporter cell lines (Fig. 5c). We quantitatively captured the transcriptional activity by measuring the fraction of cells transcribing, as determined by the presence of a detectable nascent transcription region.

Using this metric, we found significantly less transcription activity for ANKRD1 and AREG in dense cultures (Fig. 5d, e). We also tracked transcriptional activity in serum-starved cells before and after stimulation with serum. We found an immediate burst of ANKRD1 and AREG transcription following serum stimulation (Fig. 5f, g and Supplementary Movie 10). Interestingly the activity rapidly decayed to baseline levels. Comparison with YAP localization dynamics in serum replenishment assay (Fig. 3e)

**Fig. 5 Transcription dynamics of YAP target genes. a** Cartoon showing strategy for knocking in an MS2 transcription reporter cassette at the end of the coding sequence of *ANKRD1* gene. **b** Representative image of MCF10A$^{AREG}$-MS2-KI cells co-expressing MCP-mNeon with Quasar® 670-labeled RNA FISH probes against native *AREG* transcript. Repeated twice with consistent co-localization. **c** Representative images of sparse and dense cultures of MCF10A$^{AREG}$-MS2-KI showing nascent transcription spots. **d**, **e** Quantification of nascent transcription spots in sparse and dense MCF10A$^{AREG}$-MS2-KI and MCF10A$^{ANKRD1}$-MS2-KI cells. Transcription activity is the fraction of cells with a detectable transcribing locus. A one-sided Mann–Whitney *U*-test was used to test for differences in activity. **d** *ANKRD1*: $P = 0.004$, ranksum $= 40$. $N = 5$ ROI's from two independent samples per condition. **e** *AREG*: $P = 0.0079$, ranksum $= 35$. $N = 5$ ROI's from two independent samples per condition. Box and whisker lines: black-center line is median value, box-bounds are 25th and 75th percentiles, and whiskers are 0.05 and 0.95 percentiles. **f**, **g** Transcription activity of MCF10A$^{ANKRD1}$-MS2-KI **f** and MCF10A$^{AREG}$-MS2-KI **g** cell lines in serum starved cultures before and after serum stimulation. Solid line is mean of three independent experiments, shaded line is SEM. **h** Representative time-lapse projection of 10-frames before serum-starvation and 10-frames after 2 h of serum stimulation of MCF10A$^{ANKRD1}$-MS2-KI cells **i** Quantification of the number of cytoplasmic and membrane localized *ANKRD1* transcripts per cell before serum, and at various time points over a 3-h period post serum stimulation. Data for each condition was collected over a time window of 40 min, with individual data points representing the mean number of transcripts per cell for each image acquisition. A one-sided Mann–Whitney *U*-test was used to test differences in transcript counts. Preserum vs. [0–40 min]: $P = 0.85$, Ranksum $= 803$. Preserum vs. [45–85 min]: $P = 0.46$, Ranksum $= 736$. Preserum vs. [90–130 min]: $P = 1.77 \times 10^{-6}$, Ranksum $= 474$. Preserum vs. [135–175 min]: $P = 1.88 \times 10^{-8}$, Ranksum $= 424$. $N = 27$ for each condition. Data compiled from the three independent experiments in **f**. Box and whisker lines: black-center line is median value, box-bounds are 25th and 75th percentiles, and whiskers are 0.05 and 0.95 percentiles.

showed that the surge in YAP dependent transcription correlated with the early phase of YAP re-entry to the nucleus and not the longer time-scale enrichment of nuclear YAP. We further observed a steady accumulation of membrane-localized *ANKRD1* transcripts following serum stimulation (Fig. 5h and Supplementary Movie 11). Quantification of transcript density showed a significant increase in the number of membrane localized transcripts after ~2 h, with continued increase in transcript numbers even after ~3 h (Fig. 5i). These results are highly consistent with previous work, indicating our transcriptional reporter lines recapitulate known transcriptional responses of YAP. However, our ability to track and quantify the response in terms of spatiotemporal dynamics at the single-cell level alludes to a more complex picture of YAP-dependent transcription.

To test the impact of YAP localization-reset on transcription, we used TG treatment to provide a temporally synchronized input to YAP signaling. We hypothesized that the synchronized YAP localization reset following TG treatment would alter the dynamics of YAP-regulated gene transcription. Interestingly, we found that TG treatment rapidly increased both the number of cells transcribing *ANKRD1* and *AREG* (Fig. 6a, b and Supplementary Movie 12) and the intensity of nascent spots (Supplementary Fig. 5a). The calcium induced localization-reset of YAP and concomitant upregulation of YAP dependent transcription, point towards a nonlinear relationship between YAP localization and target gene activation. This is in contrast to the standard model equating nuclear enrichment with YAP activity. To test the potential connection between localization-resets and transcription, we blunted the TG-induced localization-resets with Go6976 pretreatment, which severely attenuated both *AREG* and *ANKRD1* transcriptional responses in terms of fraction of transcribing cells (Fig. 6c, d). Further, we found no significant increase in transcription pulse intensity or duration (Supplementary Fig. 5a, b). To further test a connection between YAP localization-resets and native transcriptional responses, we hypothesized that the Src inhibitor PP1 would also increase YAP responsive transcriptional outputs, based on PP1's ability to increase YAP N/C fluctuations. Indeed, both *ANKRD1* and *AREG* showed moderate increase in transcription following 1 μM PP1 treatment (Fig. 6e, f). PP1 treatment led to an immediate (albeit moderate) increase in *ANKRD1* while *AREG* increased more slowly peaking ~7 h post-treatment (Fig. 6e, f). Additionally, PP1 treatment increased the transcription pulse intensity of *ANKRD1* (Supplementary Fig. 5a) and the number of long-lived transcription pulses for *AREG* (Supplementary Fig. 5b).

To confirm that the genomic knock-in of the MS2 reporter cassette did not alter the transcriptional activity of our genes of interest, we treated wildtype MCF10A cells with TG and probed for native *AREG* transcripts using single-molecule RNA (smRNA) FISH. Indeed, we found a substantial increase in the fraction of actively transcribing loci and spot-intensity after 30 min of TG treatment, consistent with the results from MCF10A$^{AREG}$-MS2-KI (Fig. 6g–j).

Because changes in intracellular calcium levels affect many signaling effectors, we asked to what extent was the transcriptional response specific to YAP. To address this, we performed smRNA FISH against the commonly used housekeeping gene glyceraldehyde 3-phosphate dehydrogenase (*GAPDH*). We found a minor increase in transcription spot intensity after TG treatment, but not a significant increase in the fraction of cells actively transcribing. When compared to *AREG* smRNA FISH, the *GAPDH* response is minimal, suggesting the transient SOCE specifically affects YAP target genes at short time scales.

We expanded this measurement to assay global mRNA expression using RT-qPCR against a host of YAP target genes (*AREG, ANKRD1, IGFBP3, TGFB2, CTGF, CYR61, FOXF2, RASSF2*)[58] and housekeeping genes (*GAPDH, PGK1, HPRT*) (see "Methods" section, Supplementary Table 3). Consistent with our nascent transcription kinetics data and smRNA FISH data, we found that YAP gene expression generally increased following TG treatment, while housekeeping genes did not show any appreciable change in transcription (Fig. 6m). This result both validates our transcription kinetics data and demonstrates a specific regulation of YAP target genes.

To test if a fundamental connection exists between YAP shuttling and the transcription of YAP-responsive genes in the context of native cellular processes, we investigated mitosis, since mitotic chromatin condensation has been deemed as a major mediator of TF displacement from bulk chromatin[59]. Recent work has provided evidence for regulation of Hippo signaling by processes intrinsic to cell cycle progression[60]. We observed that YAP was excluded from condensed chromosomes (Fig. 7a) during mitosis, suggesting that localization-resets are intrinsic to cell division. Might nuclear re-entry of YAP upon mitotic exit also reactivate YAP-dependent transcription? We approached this question with the real-time transcription reporter lines, by synchronizing cultures to the G1/S phase boundary using a double thymidine block and imaging continuously for 24 h upon release of the block. Aligning the single cell transcription traces by cytokinesis revealed that both *AREG* and *ANKRD1* transcription peaked over a 2-h period following mitotic exit (Fig. 7b–e and Supplementary Movie 13) before gradually decaying over the next 16 h (Fig. 7d, e). This remarkable pattern suggests that target-gene expression activates shortly after a YAP localization-reset

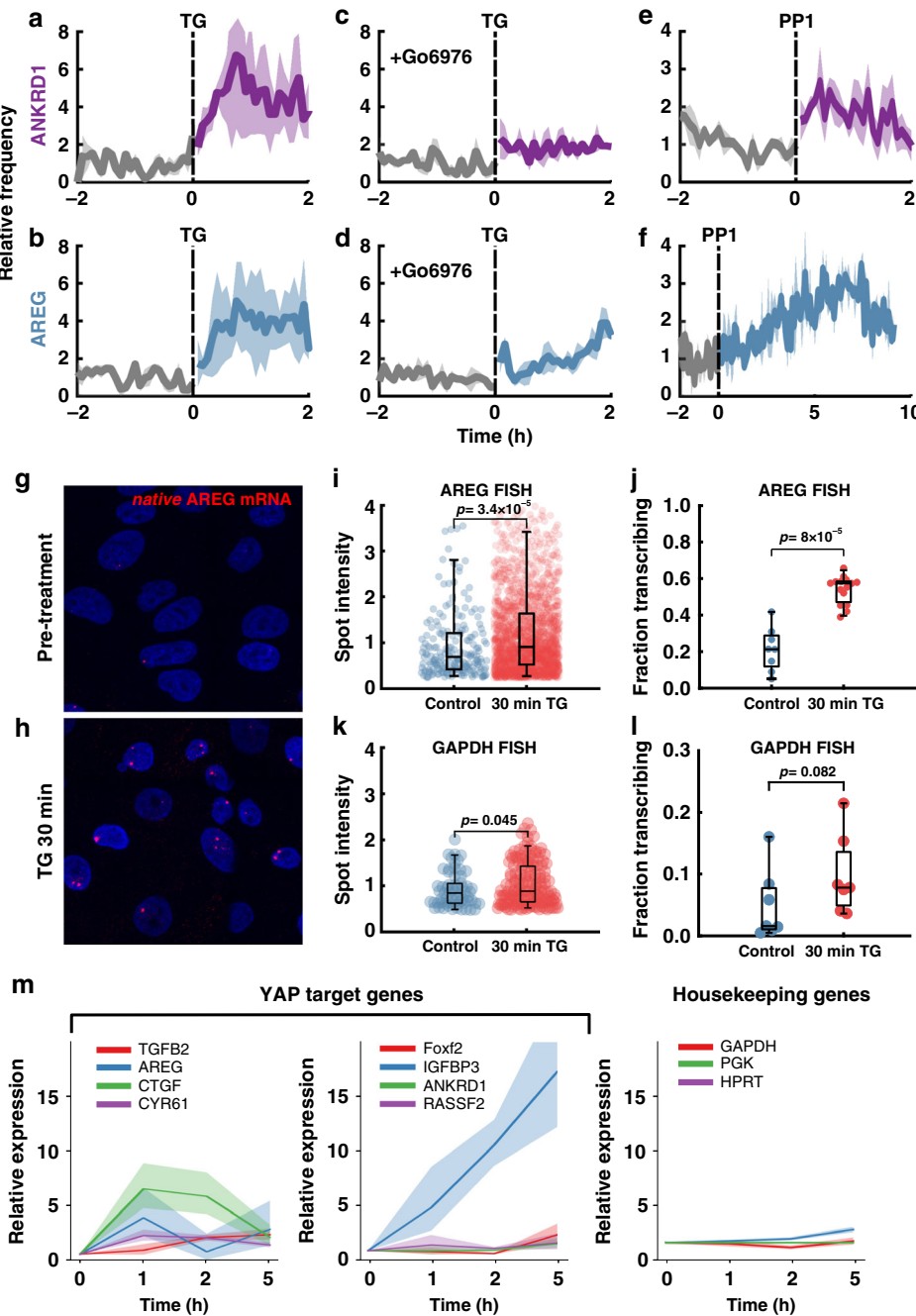

**Fig. 6 Transcription dynamics of YAP target genes during localization-reset. a–f** Mean transcriptional frequency before and after drug treatment. Transcriptional frequency was normalized to 1 for the 2-h pretreatment period for all data shown in **a–f**. **a, b** Treatment with TG, mean $+/-$STD of three experiments for *ANKRD1* **a** and *AREG* **b**. **c, d** Pretreatment with 1 μM Go6976, followed by TG, mean $+/-$STD of three experiments for ANKRD1 (**c**) and *AREG* (**d**). **e, f** Treatment with 1 μM PP1, mean $+/-$STD of 2 (*ANKRD1*) (**e**) and 3 (*AREG*) (**f**) experiments. **g, h** Representative smRNA FISH against native *AREG* transcripts before and 30 min after TG treatment. **i–k** Quantification of *AREG* and *GAPDH* FISH before and after TG treatment. Statistical testing was done using the one-sided Mann–Whitney $U$-test. **i** $P = 3.4 \times 10^{-5}$, Ranksum = 176,838. $N_{control} = 198$, $N_{TG} = 1916$ spots. **j** $P = 8 \times 10^{-5}$, Ranksum = 37. $N_{control} = 8$, $N_{TG} = 15$ ROIs. **k** $P = 0.045$, Ranksum = 6208. $N_{control} = 63$, $N_{TG} = 156$. **l** $P = 0.082$, Ranksum = 41. $N_{control} = 7$, $N_{TG} = 7$ ROIs. ROIs were separate acquisitions of different regions on a sample, containing tens of cells each. Data compiled from two independent experiments per condition. Box and whisker lines: black-center line is median value, box-bounds are 25th and 75th percentiles, and whiskers are 0.05 and 0.95 percentiles. **m** Time-course of expression of YAP target genes (*TGB2, AREG, CTGF, CYR61, Foxf2, IGFBP3, ANKRD1, RASSF2*) and housekeeping genes (*GAPDH, PGK1, HPRT1*) before and after TG treatment using RT-qPCR. Solid lines are means and error envelopes are standard error of Mean obtained from three biological replicates and three technical replicate for each biological replicate.

and is followed by a currently undiscovered intrinsic inhibitory process despite YAP remaining nuclear. The complex activation and temporal dynamics of YAP-dependent transcription highlights that transcription levels cannot by inferred by a simple nuclear abundance metric for YAP. This temporal pattern does indeed coincide with previous measurements showing that the APC/C$^{Cdh1}$ complex degrades LATS1/2 in G1 phase, allowing increased YAP-dependent transcription[60].

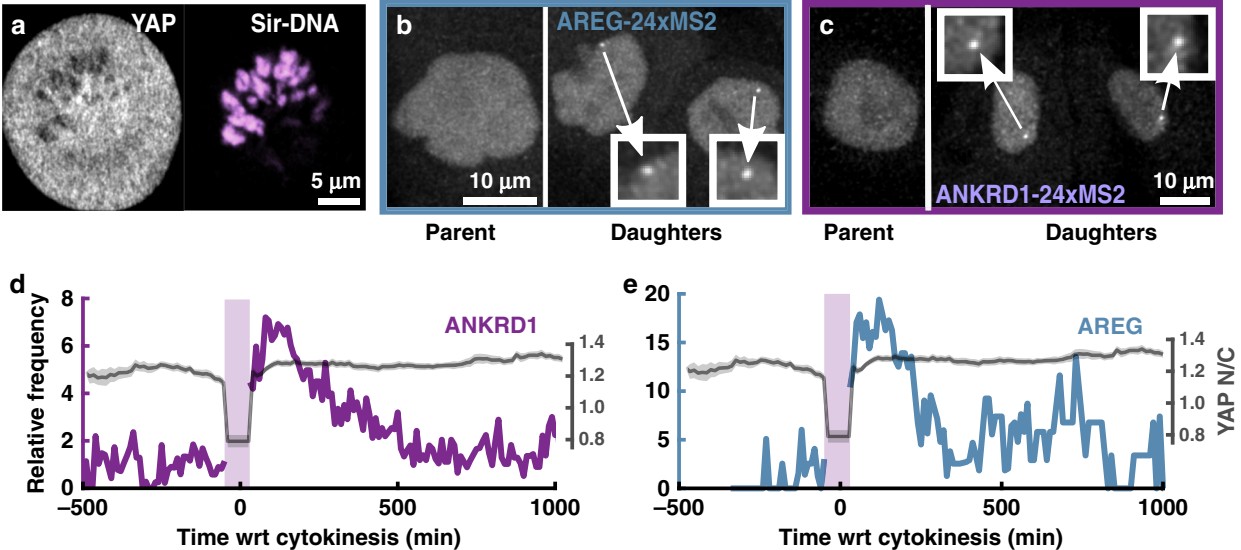

**Fig. 7 YAP localization-reset and transcription activation following mitosis. a** Representative YAP localization (gray) in a dividing cell (condensed chromatin shown in magenta). Repeated twice with similar results in all cells. **b**, **c** Example of MCF10A$^{AREG\text{-MS2-KI}}$ **b** and MCF10A$^{ANKRD1\text{-MS2-KI}}$ **c** cells going through mitosis. Insets show nascent transcription spots in both daughter cells. **d**, **e** Left axis: mean transcription frequency of cells showing transcription reset after mitosis for *ANKRD1* (**d**) and *AREG* (**e**). Mitotic events collected from two independent experiments for both *ANKRD1* ($N = 39$ divisions) and *AREG* ($N = 25$ divisions). Right axis: mean YAP N/C during division ($N = 24$ divisions). The N/C ratio during mitosis was calculated as the mean value from seven high-magnification images of mitotic cells expressing YAP (as shown in **a**).

**Enhanced nuclear retention of YAP is anticorrelated with YAP activity**. Finally, we investigated how localization-resets are affected by oncogenic transformation, which permanently alters YAP regulatory circuits[8]. We found that transformation dampened localization-resets (Fig. 3c, d) and eliminated cell-density sensing (Fig. 2c–e) while increasing transcription output of commonly assayed YAP target genes (Supplementary Fig. 6). Although the YAP target CTGF was decreased after Ras transformation in MCF10A cells, increased expression is seen after HRas$^{G12V}$ expression in HEK293T cells[25]. To further explore the relationships of YAP localization control and transcription in an oncogenic context, we generated YAP-eGFP knock-ins on SUM159 and MDA-MB-231, two patient-derived lines that harbor a host of oncogenic mutations, including constitutively active Ras (H-Ras$^{G12V}$ in SUM159 and K-Ras$^{G13D}$ in MDA-MB-231). All of these transformed cell lines (MCF10AT$^{YAP\text{-GFP-KI}}$, SUM159$^{YAP\text{-GFP-KI}}$, and MDA-MB-231$^{YAP\text{-GFP-KI}}$) showed reduced YAP localization response upon TG treatment (Fig. 8a), suggesting that oncogenic transformation perturbs YAP localization control. This is counterintuitive, since, if localization-resets drive YAP target gene transcription, why are YAP responsive genes chronically upregulated in transformed cells that lack localization-resets?

Perhaps, the chronically elevated transcriptional regulation in transformed cells operates through the same mechanism, however under different steady-state transport conditions. Although our results highlight a strict connection between concerted localization dynamics of YAP (i.e., most molecules leaving and re-entering the nucleus together) and transcription, we must consider that cells at steady-state will have a constant, stochastic transport of individual YAP molecules between the nucleus and cytoplasm. These transport rates could be tuned to regulate the baseline transcription of YAP genes.

To measure first-order nuclear import and export rate of YAP, we carried out fluorescence recovery after photobleaching experiments, where we monitored eGFP fluorescence in the nucleus after bleaching either the nuclear or cytoplasmic signal, in cells expressing native YAP-eGFP or an inert nuclear transport cargo (NLS-eGFP$_{2X}$-NES) (Fig. 8b). We found that the three transformed lines had upregulated YAP nuclear export and import rates compared to MCF10A (Fig. 8b, c). However, MCF10A had the highest ratio of import to export ($\bar{k}_{import}/\bar{k}_{export}$ = 3.9) compared to all transformed cell lines (H-Ras = 1.7, SUM159 = 1.7, and MDA-MB-231 = 1.6). The upregulated import and export rates of YAP in transformed cell lines imply that YAP rapidly equilibrates between the nucleus and cytoplasm. Using an engineered nuclear transport reporter that harbors a nuclear localization signal and a nuclear export signal, we found that the base-line (i.e., non-YAP) nuclear export rates across all cell lines were not significantly different (Fig. 8c), suggesting that YAP transport rates are specifically affected by oncogenic transformation.

Previous work has shown that YAP has increased interactions with chromatin in cancer-associated fibroblasts[16]. To understand whether malignant transformation affects YAP-chromatin interactions in breast epithelial cells, we performed high time-resolution spatiotemporal FRAP experiments on native YAP-eGFP in the nucleus. Briefly, a 2 pixel wide line bisecting a circular bleach spot was scanned repeatedly at high speed (~220 Hz) to generate radial intensity profiles of fluorescence recovery at a high temporal frequency. This allowed for more accurate fitting than in case of conventional FRAP, which is critical for binding and diffusivity analysis of fast exchanging TFs[61–63]. Inspection of the raw FRAP data (time vs. intensity carpet), showed that MCF10A cells recovered fluorescence on a much longer time-scale than all three transformed lines (Fig. 8d), indicating a potentially stark change in chromatin interactions with YAP. Indeed, after fitting the spatiotemporal data to either pure diffusion or diffusion–reaction models[62] (Supplementary Fig. 7 and Supplementary Table 4), we found striking differences in YAP binding kinetics. In MCF10A cells, FRAP data fit well to a diffusion–reaction model with a single binding mode (Supplementary Fig. 7 and Supplementary Table 4). We measured an effective binding equilibrium close to 1, such that at any time point, approximately 50% of YAP molecules were effectively bound. Because the equilibrium constant is close to 1, this bulk

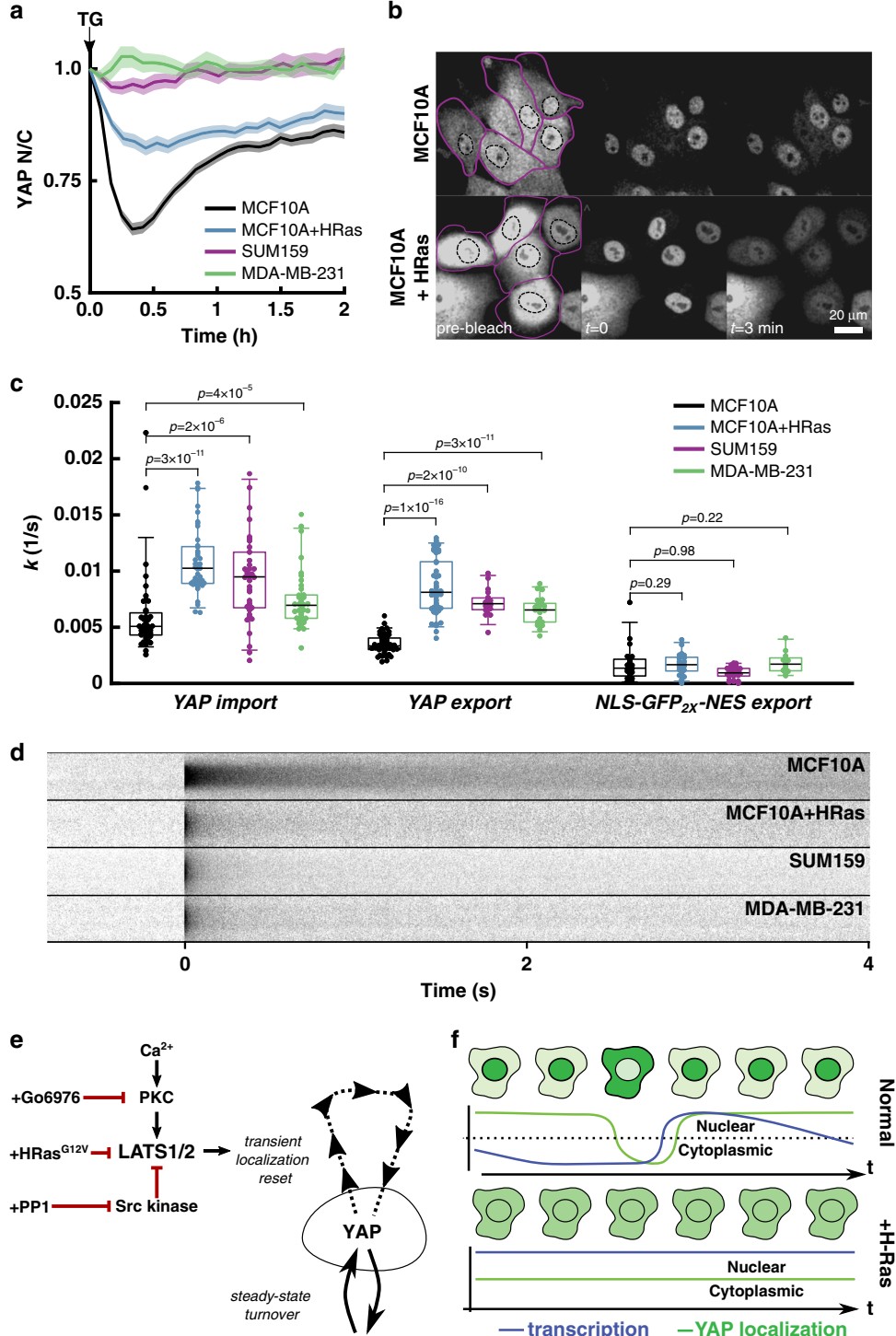

interaction mode is relatively weak, but it is globally present throughout the nucleus.

Consistent with the rapid FRAP recovery of YAP in the nucleus (Fig. 8d), all three oncogenically transformed lines were best described by a pure diffusion model (Supplementary Fig. 7 and Supplementary Table 4). These results suggest that bulk interaction in the nucleus, most likely to chromatin, could restrain YAP nucleocytoplasmic transport in non-transformed cells through an unidentified binding partner. Whereas transformation abolishes this interaction, which could either be specific to Ras activation or a more general oncogenic signaling state.

## Discussion

The currently prevailing model of the relationship of YAP localization to transcription is linear, with more nuclear YAP being equated with more transcription. Through direct imaging and quantification, we highlight the utility of genomic reporter knock-ins for monitoring native TF localization and activity. Having access to single-cell dynamics allowed us to develop a deeper understanding between YAP localization and downstream transcription. We present evidence that active transcription is not simply linearly correlated to nuclear YAP levels under a variety of signaling contexts.

**Fig. 8 Ras transformation alters YAP dynamics. a** Mean YAP $N/C$ after TG treatment for MCF10A ($N = 157$ tracks), MCF10A + HRas ($N = 75$ tracks), SUM159 ($N = 70$ tracks), MDA-MB-231 ($N = 101$ tracks). Data is compiled from two independent experiments for each condition. Solid line is mean and shaded area is SEM. **b** Representative images of FRAP experiments where cytoplasmic pool of YAP-eGFP is bleached and the nuclear pool is tracked over time. Nuclear boundaries are drawn with dashed black line while cytoplasm boundaries are drawn with solid purple line. HRas transformed cells show significant re-distribution by 3 min compared to MCF10A control. **c** Distribution of specific import and export rates of YAP, and the export rate of NLS-eGFP2x-NES for MCF10A ($N_{YAP\text{-}imp} = 43$, $N_{YAP\text{-}exp} = 49$, $N_{exp} = 21$), MCF10A + HRas ($N_{YAP\text{-}imp} = 41$, $N_{YAP\text{-}exp} = 46$, $N_{exp} = 21$), SUM159 ($N_{YAP\text{-}imp} = 38$, $N_{YAP\text{-}exp} = 19$, $N_{exp} = 27$), and MDA-MB-231 ($N_{YAP\text{-}imp} = 38$, $N_{YAP\text{-}exp} = 23$, $N_{exp} = 12$) YAP-GFP knockin cell lines. Data aggregated from at least two independent cultures for each cell line. Box and whisker lines: black-center line is median value, box-bounds are 25th and 75th percentiles, and whiskers are 0.05 and 0.95 percentiles. $P$-values are calculated from one-sided Mann–Whitney $U$-test to test whether import/export was changed from baseline MCF10A. YAP-import: +HRas ($P = 3 \times 10^{-11}$, Ranksum = 1093), SUM159 ($P = 2 \times 10^{-6}$, Ranksum = 1270), MDA-MB-231 ($P = 4 \times 10^{-5}$, Ranksum = 1348). YAP-export: +HRas ($P = 1 \times 10^{-16}$, Ranksum = 1249), SUM159 ($P = 2 \times 10^{-10}$, Ranksum = 1231), MDA-MB-213 ($P = 3.2 \times 10^{-11}$, Ranksum = 1247). Baseline export: +HRas ($P = 0.29$, Ranksum = 429), SUM159 ($P = 0.98$, Ranksum = 615), MDA-MB-231 ($P = 0.2$, Ranksum = 336). **d** Carpet plot generated from the average line FRAP of YAP-eGFP in various cell lines. Data compiled from at least two independent experiments per condition. **e** Summary of induction and modulation of transient YAP localization-resets through LATS kinase. **f** Proposed model for the relationship between YAP dynamics and YAP responsive transcription. Dashed line represents $N/C = 1$.

When we extended our findings to transformed cell lines, we directly observed lower levels of bulk YAP binding in the nucleus being associated with higher YAP activity. Combining the single cell studies of normal and transformed cells, we propose a model of transcription whereby prolonged nuclear retention causes transcriptional inhibition. In this model, the enhanced chromatin interactions may constitute a nonspecific retention mechanism that inhibits YAP transcriptional output. The localization-resets oppose this inhibition by transient activation of YAP. On the other hand, Ras-transformation abolishes nuclear retention, allowing escape from compartmentalization-based control through a rapid nucleocytoplasmic exchange (Fig. 8e). Previous work suggests that the increased activity of YAP in Ras-transformed pancreatic cells reflects increased YAP levels[24]. Our observations raise the possibility that amplification of YAP-dependent transcription is not merely due to increased expression levels but also due to dysregulated nucleocytoplasmic dynamics.

It has been suggested that nuclear retention can inhibit TF activity through posttranslational modifications, which can be reversed through re-localization to the cytoplasmic compartment, as seen in nuclear factor erythroid 2-related factor 2 (Nrf2)[52]. This raises the intriguing possibility that prolonged nuclear retention deactivates YAP, whereas Ras transformation maintains YAP in a hyperactive form through rapid nucleocytoplasmic turnover (Fig. 4f). Modulation of YAP activity in a variety of cellular contexts is underscored by multiple post-translational modifications (PTMs). For example, phosphorylation of nuclear YAP at Y357 reduces its transcriptional competence without affecting its localization[14]. Potential mechanisms behind the control of nuclear YAP activity include nuclear phosphorylation at Y357 and the specific inhibition of YAP through the SWI/SNF complex[53]. Further exploration of compartment-specific PTMs and binding partners may offer further insight into the mechanisms behind dynamic transcription control through YAP.

While our results demonstrate a strong correlation between YAP localization-reset and gene expression, an important next step would be to develop a live imaging platform to establish the temporal relationship between YAP localization fluctuation and transcriptional activation at single-cell resolution. Transcriptional pulses are regulated combinatorially, where the response is dependent on the appropriate levels and localization of multiple effectors. For example, in the case of the β-actin gene, the appropriate transcriptional response to serum-induced signaling is dependent on the right balance of serum response factor and actin[64]. MAL, a monomeric actin-binding protein, that shuttles between nucleus and cytoplasm, accumulates rapidly in the nucleus upon serum stimulation[65], similar to the rapid shift in YAP localization upon serum starvation and replenishment.

Identifying molecular partners that can fine-tune YAP localization resets in a context-specific manner is an important future direction. While oscillations in localization of many transcription factors such as NFAT, NFkB, p53, and Wnt, have been connected to transcriptional response dynamics in a wide variety of cellular contexts[11–13,66–68], one future challenge will be to assess how differences in magnitude and frequency of fluctuations in otherwise phenotypically identical cells could potentially affect cellular fate. Fluctuations in gene expression are known to alter cell-fate decisions[69], and YAP has been implicated in stem cell pluripotency[70]. Transient reversals in YAP localization acting as a fast trigger of YAP activity could be relevant for spontaneous cell fate switching in stem cells as well as lineage reprogramming[71,72]. Combining real-time detection of dynamic fluctuations in YAP localization with single-cell genomics[73,74] and lineage tracing[75,76] will help to identify the long term effects of YAP fluctuation on cell fate decisions.

Our data show a transient but robust spike in the transcription of select YAP target genes upon mitotic exit, suggesting a possible role of YAP in mitotic bookmarking, which is a fast reboot mechanism for the expression of genes involved in metabolism and cell survival following cytokinesis[77–79]. While YAP itself shows marked exclusion from condensed chromatin during mitosis, the effective YAP localization reset during early stages of mitotic exit, may initiate new rounds of engagement between transcriptionally competent YAP and chromatin localized TEAD. Whether YAP is a dynamic bookmarking factor that promotes postmitotic expression of YAP target genes across the board, needs to be explored, especially in the context of stem cell renewal and embryonic development.

## Methods

**Cell culture**. *MCF10A, MCF10AT were cultured as previously described*[80]: Briefly cells were grown in DMEM/F12 + 5% Horse serum supplemented with EGF (20 ng/ml), Insulin (10 µg/ml), Cholera toxin (100 ng/ml), Hydrocortisone (0.5 µg/ml) and Pen/Strep[80]. Cells were passaged every third day. SUM159, MDA-MB-231 were cultured in DMEM+ 10% FBS + Pen/Strep. All cell lines were maintained at 37 °C and 5% $CO_2$ either in tissue culture incubators or microscope incubators.

For H1 hESC cells, prior to starting cell culture, tissue culture flasks or MATTEK glass bottom dishes were incubated in sufficient volume of a mixture of Essential 8 (E8) media (ThermoFisher cat. # A1517001) and Matrigel (Corning cat. #354277) mixture (9:1, Matrigel:E8 media) for 1 h at room temperature (RT). H1 human embryonic stem cells were cultured in E8 media at 37 °C, 5% $CO_2$, with fresh media added daily, and were passaged when cells reached 80–90% confluence.

All live-cell imaging assays that required tracking of cells were performed using SiR-DNA™ nuclear stain (Cytoskeleton, inc.). SiR-DNA™ was added at 0.5 µM for 1 h, followed by one wash with warm media.

**Acrylamide hydrogel for seeding cells**. Acrylamide hydrogel was prepared according to published protocols[81]. Briefly, glass-bottomed dishes were first cleaned and activated with 1% v/v (3-Aminopropyl) triethoxysilane (APTES,

Sigma–Aldrich) in ethanol. The silanized coverslips were baked at 65 °C overnight. Next the glass surface was treated with 0.5% v/v glutaraldehyde in Milli-Q water for 30 min, then cleaned and dried. The gel solution (acrylamide, bis-acrylamide) in phosphate buffered saline (PBS) was first degassed in a vacuum desiccator for 1 h. After that, 10 μl of 10% w/v ammonium persulfate (APS, Sigma–Aldrich) and 1 μl of $N,N,N',N'$-Tetra-methyl-ethylenediamine accelerator (TEMED, Sigma–Aldrich) per ml of solution was added. After mixing, the solution was poured onto the glass top and covered with a coverslip. After 30 min of incubation at RT, coverslips were removed, and dishes were washed three times with PBS. The gel was then incubated with 0.5% Sulfo-Sanpah (Fisher) in water and exposed to UV light (UV cross linker) for 10 min. After repeating incubation and UV exposure, the dishes were washed and incubated in a fibronectin solution (0.1 mg/ml) overnight. Cells were seeded the next day after thoroughly cleaning the dishes.

**Cell culture treatments**. *Thapsigargin (TG)*: Alfa Aesar—#J62866. One millimolar DMSO stock. Cells were treated at 1 μM and imaged immediately.

*Src inhibitor (PP1)*: Cayman Chemical Company—#14244. Ten millimolar DMSO stock. Cells were treated at 10 μM and imaged after 1 hr for YAP fluctuation experiments or immediately for transcription experiments.

*PKC inhibitor (Go 6976)*: Tocris, #2253. One millimolar DMSO stock. Cells were treated at 1 μM for 2 h before being treated by TG.

*Serum starvation*: MCF10A cells were seeded on MATTEK dishes (35 mm, No. 1.5 glass) and grown in complete media for 3 days before serum starvation in DMEM/F12 without other supplements for 20 h. These cells were then either replenished with serum or treated with 1 μM lysophosphatidic acid (Santa Cruz Biotechnology).

**Capillary western blot**. Cells grown to ~80% confluence were trypsinized, washed once with PBS, pelleted and flash frozen. Raybiotech (Peachtree Corners, GA) Auto-Western Service was used to carry out capillary western blot using in-house antieGFP antibody and antiGAPDH antibody provided by Ray-Biotech. Western blot was performed by using an automated Capillary Electrophoresis Immunoassay machine (WES™, ProteinSimple Santa Clara, CA).

**Cell line generation and endogenous gene tagging using CRISPR-Cas9**. For C terminal tagging of YAP and TEAD with GFP/mCherry we generated donor plasmids for homology dependent repair (HDR), where the general design of the donor plasmid consisted of an upstream homology arm (~1 Kb long) followed by GFP/mCherry-(P2A-Puromycin/Hygromycin-stop codon) cassette followed by a downstream homology arm (~1 Kb long). MCF10A cells grown to ~80% confluence was trypsinized and electroporated with the donor plasmid, guide RNA plasmid and a plasmid expressing SpCas9 at a ratio of 2:1:1 (total 12 μg) using Neon electroporator (Life Technologies) and a 30 ms:1100 V:2 pulse electroporation program. Following electroporation cells were grown for three days before initiating antibiotic selection. For antibiotic selection, fresh media containing 1 μg/ml Puromycin or 250 μg/ml Hygromycin was added to the cells every two days. Post selection cells were grown in antibiotic-free media. For both YAP and TEAD, tagging efficiency was nearly 100% as nearly all cells post-selection showed appropriate localization of the FP tagged proteins and were genomically stable over at least 20 division cycles. For further validation genomic sequences containing the knockins were PCR amplified. For generating YAP GFP knockins of MCF10AT, SUM159, MDA-MB231, and H1 hESCs, cells were electroporated using a Neon electroporator then selected as described above. For MDA-MB231 a 10 ms:1400 V:4 pulse electroporation program was used, for MCF10AT and SUM159 we used a 30 ms:1100 V:2 pulse electroporation program and for H1 hESC a 30 ms:1050 V:2 pulse electroporation program was used. Specifically, to generate the H1 hESC YAP-eGFP cell line, cells were grown to ~70–80% confluence in a T-75 flask, dislodged using EDTA as described above, washed in PBS and then electroporated with plasmids. Electroporated cells were seeded in fresh E8 media containing 5 μM Y-27632 dihydrochloride (a selective ROCK inhibitor, Tocris) and allowed to recover for 48 h before antibiotic selection.

For generating cell lines that can report on native transcription kinetics of YAP responsive genes, MCF10A cells were first transfected with a Super Piggy BAC Transposase expression vector (SBI) and a custom generated PiggyBAC vector carrying an MCP-mNeon gene driven by TRE3G promoter and a rTTA3 (tetracycline-controlled transactivator 3)-T2A-Hygromycin cassette driven by a PGK promoter, followed by selection with 250 μg/ml hygromycin. To insert a 24× MS2 transcription reporter cassette at the beginning of the 3′ UTR of *AREG*, we generated a donor plasmid for homology dependent knockin, where the general design consisted of an upstream homology arm (~1 Kb long) followed by a HA tag-P2A-Blasticidine-stop codon-24× MS2 cDNA cassette followed by a downstream homology arm (~1 Kb long). For *ANKRD1*, which had a low CRISPR mediated knockin efficiency, we used a double cut HDR donor plasmid where the "homology-knockin" cassette was flanked by single guide RNA (sgRNA)-PAM sequence on either side. This greatly increased knockin efficiency for *ANKRD1*. Cells were selected with 10 μg/ml Blasticdine.

For calcium sensing, cells were transduced with lentiviral particles reconstituted from a lentiviral vector expressing GcAMP6f from a constitutively active CMV promoter.

To minimize any potential confounding effect of differences in source of origin of MCF10A and MCF10AT, we generated a constitutively active H-Ras (H-Ras$^{G12V}$) transformed MCF10A$^{YAP-GFP-KI}$ cell line using lentiviral transduction and subsequent selection for neomycin resistance using Geneticin (400 μg/ml). This cell line was used specifically for the line-FRAP, and import/export measurements.

Details of cell lines.

| Cell line | Source cell line | Method of cell line generation | Antibiotic resistance |
|---|---|---|---|
| MCF 10A YAP-eGFP | MCF10A | CRISPR knock in | Puromycin |
| MCF 10AT YAP-eGFP | MCF10AT | CRISPR knock in | Puromycin |
| MCF 10AT YAP-eGFP | MCF10A YAP-eGFP | CRISPR knock in +HRas Lentiviral transduction | Neomycin |
| SUM159 YAP-eGFP | SUM159 | CRISPR knock in | Puromycin |
| MDA MB 231 YAP-eGFP | MDA MB 231 | CRISPR knock in | Puromycin |
| H1 hESC YAP-eGFP | H1 hESC | CRISPR knock in | Puromycin |
| MCF 10A YAP-eGFP +TEAD1 mCherry | MCF10A | Dual CRISPR knock in | Puromycin + Hygromycin |
| MCF 10A MCP mNeon | MCF10A | PiggyBAC transposition | Hygromycin |
| MCF 10A MCP mNeon *AREG* 24X MS2 | MCF10A MCP mNeon | CRISPR knock in | Hygromycin + Blasticidin |
| MCF 10A MCP mNeon *ANKRD1* 24X MS2 | MCF10A MCP mNeon | CRISPR knock in | Hygromycin + Blasticidin |
| MCF 10A GcAMP6f | MCF 10A | Lenti viral transduction | Puromycin |
| MCF10A YAP-eGFP +LAM Δ50 mCherry | MCF 10A YAP-eGFP | PiggyBAC transposition | Hygromycin |
| MCF 10A NLS-GFP$_{2X}$-NES | MCF 10A | Transient transfection | |
| MCF 10AT NLS-GFP$_{2X}$-NES | MCF 10AT | Transient transfection | |
| SUM 159 NLS-GFP$_{2X}$-NES | SUM 159 | Transient transfection | |
| MDA MB231 NLS-GFP$_{2X}$-NES | MDA MB231 | Transient transfection | |

For guide sequences used for CRISPR knockin see Supplementary Table 1.

**Genomic PCR**. All crispr knockin cell lines were validated using genomic PCR, with primers located in the upstream and downstream homologous recombination arms. Primers and amplicon sizes are provided Supplementary Table 2.

**Immunofluorescence**. MCF10A$^{YAP-GFP-KI}$, MCF10A, and MCF10A$^{LATS1/2\ KO}$ cells were washed in PBS and then fixed in 4% paraformaldehyde in PBS for 10 min at room temperature. Fixed cells were washed thrice in PBS and then permeabilized in PBS containing 0.1% (v/v) Triton X-100 for 10 min, followed by another three rounds of washing in PBS. Permeabilized cells were then incubated with a blocking buffer containing 1% BSA, 22.52 mg/ml glycine, and 0.1% Tween 20 in PBS for 1 h (PBST). Cells were then incubated with anti YAP Antibody (63.7) (Santa Cruz biotechnology: Catalog # sc-101199) in 1% BSA in PBST overnight at 4 °C at 50× dilution, followed by three 5 min washes with PBS. Cells were then incubated with Goat antiMouse IgG2a Cross-Adsorbed Secondary Antibody, Alexa Fluor 555 (Thermo Fisher Scientific: Catalog # A-21137) at 10μg/ml concentration in PBS containing 1% BSA. In the final step cells were washed thrice in PBS and counterstained with Hoechst nuclear stain.

**Microscopy**. Live imaging was done on either Zeiss LSM700 or an Olympus FV10i. Cells plated on fibronectin-coated MATTEK dishes (35 mm, No. 1.5 glass) were imaged either with 63×/1.40 oil immersion (Zeiss) or 60×/1.2 water immersion (Olympus) objective. All multi-day image acquisitions were carried out on Olympus FV10i. All photo-bleaching experiments were performed on Zeiss LSM700 using the following parameters:

| Photobleaching experiments | Laser (nm) | Frame rate | # Prescan frames | # Bleach scans | # Postscan frames |
|---|---|---|---|---|---|
| Import/export | 488 | 10 s | 1 | 12 | 12 |
| YAP spatiotemporal FRAP | 488 | 4.6 ms | 500 | 8 | 1000 |

**Image analysis**. All calculations and analysis were performed using MATLAB, Python, and ImageJ. The cell/spot segmentation, tracking, and analysis functions are publicly available here: https://github.com/jmfrank/track_analyzer. FRAP model fitting was implemented in Mathematica based on Stasevich et al.[62]

**Cell tracking**. Cell tracking was performed following these steps: (1) anisotropic diffusion filtering and sum of signal derivatives, (2) applying global threshold to identify nuclei, (3) seeded watershed segmentation to breakup regions into single nuclei, (4) frame-to-frame tracking using a distance minimizing linking algorithm. Tracking was verified by eye using a GUI, and artifactual tracks were flagged and excluded by downstream analysis.

**N/C ratio**. We define the N/C ratio as the ratio of nuclear to cytoplasmic concentrations of YAP (or TEAD), assessed by the mean fluorescence intensity in the nucleus and cytoplasmic volumes after subtracting fluorescence background. The masks used to measure N/C was a 3-plane volume, centered by the plane with maximum Sir-DNA intensity. Because YAP is physically excluded from the nucleoli, we removed the nucleolar regions (determined by the lower 35th percentile of intensities within the nucleus boundary). The cytoplasmic region was determined by generating a mask extending 0.5 μm from the nucleus boundary into the cytoplasm. Because TEAD is mostly nuclear, we used the TEAD signal to segment the nucleus rather than Sir-DNA.

**Local cell density**. To account for variable density in a growing monolayer, we estimated the local cell density for each detected cell as the number of cell centroids found within a search radius of 250 pixels (69 μm) of the cell of interest centroid (Fig. 2c, e).

**YAP N/C fluctuation analysis**. Rapid changes in YAP signal were defined as signals that are continuously increasing or decreasing by at least 0.12 (N/C magnitude). If the signal did not change at least 0.005 over a 3-frame window, then the signal was considered stagnated and continuity was broken.

**Wound assay**. MCF10A cells were seeded on MATTEK imaging dishes at 50% confluency. After 3 days of growth, the very dense monolayer was imaged (Fig. 2g prewound). The dish was then gently scratched with a 20 μL pipette tip and immediately imaged. About a 1-min delay between wounding and the first image due to incubator adjustment and microscope focusing.

**GcAMP6f imaging**. Similar to the YAP signal, we measured the GcAMP6f signal as the average of a 0.5 μm wide contour extending into the cytoplasm from the nuclear boundary, as the GcAMP6f signal is mostly excluded from the nucleus.

**Single molecule RNA FISH**. Stellaris RNA FISH probes were purchased from Biosearch technologies. FISH probe hybridization was carried out as per manufacturer's protocol (https://biosearchassets.blob.core.windows.net/assets/bti_stellaris_protocol_adherent_cell.pdf). Post hybridization cells were imaged using a Zeiss LSM700.

**Nascent transcription**. Nucleus segmentation was performed in 3D to ensure each nascent transcription spot was located within the nucleus. Although we sorted MCP-mNeon expressing cells, the expression per cell was variable, requiring local thresholding to segment cells with different expression levels. This was done using the following steps:

Applying filtering (as described in the 2D Sir-DNA segmentation).
Finding local maxima.
Applying a lower-bound threshold to include all regions containing a local maxima.
Looping over all regions found in step 3, use the derivative of the intensity-percentile distribution (dI/dP) of pixel intensities to determine the local threshold. A peak in dI/dP indicates the background to foreground boundary (see plot below). With no significant peak, the region is properly segmented. Because the local threshold is not the same for all regions, one needs to look at the histogram information in terms of percentile to find the foreground vs. background for each subregion.
Applying a seeded watershed to separate connecting nuclei.

After identifying cell nuclei, actively transcribing cells were identified by looking for a sharp, bright signal (for MCP-mNeon) within the segmented nuclei. Spots were detected in two steps to ensure robust quantification:

Potential spots were first identified by using a Laplacian of Gaussian filter, followed by a local thresholding by identifying potential spots as regions of pixel intensity greater than 99.97th percentile of all pixels within a particular nucleus. An integrated intensity threshold is then applied to these potential spots from step 1. For each potential spot, a background measurement was made using a shell mask centered at the centroid of the potential spot (inner diameter 7 pixels, outer diameter 9 pixels). The integrated intensity is the sum of all pixel values within the inner shell region (i.e., pixels located less than 3.5 pixels from centroid).

Particle tracking was applied to nascent spots to gather statistics on pulse duration and intensity.

**Analysis of mitotic cells**. Mitotic cells were manually annotated using a GUI to mark the time of cytokinesis, parent cell, and daughter cells for each division. The data extracted from the cells is then aligned by the time of cytokinesis. To create the YAP N/C trace through division, the N/C trace of the parent cell was concatenated with the mean N/C values of the daughter cells. Nascent transcription through mitosis was evaluated by measuring the fraction of cells with at least one nascent transcription spot and normalizing the premitosis fraction to 1. Because the premitosis nucleus is diploid, there are 2× copies of the MS2 gene cassette and each daughter receives 1× copies. Therefore, the postmitosis fraction of cells is measured by whether either of the daughters are transcribing, which means the same number of MS2 gene cassettes are monitored before and after division.

**RT-qPCR**. Gene expression was quantified by qPCR on Lightcycler 480 (Roche). Briefly, Total RNA, extracted using Qiagen Rneasy Mini kit was used as input. Six PCR reactions including two biological replicates and three technical replicates per gene were performed using Brilliant SYBR master mix (Agilent), running cycles of 10 min at 95 °C, and 40 cycles of 20 s at 95 °C, 15 s at 54 °C, and 45 s at 72 °C. For primer sequences for each gen, see below. The Ct (crossing threshold) value per reaction, defined as the second derivative maximum (SDM) of the amplification curve was obtained using the Lightcycler built-in software. Expression value per gene was averaged from replicates. Fold of expression change was calculated by 2^ (delta Ct) and normalized to HPRT1. For further verification we also measured the expression of two other housekeeping genes GAPDH and PGK1. For a list of RT-qPCR primers, see Supplementary Table 3.

**Spatiotemporal line FRAP of nuclear YAP-eGFP**. We found that the YAP-eGFP recovery is very rapid, so we opted to use a circular ROI for bleaching (10 pixel radius, 0.9 μm) but a line ROI (2 pixel wide) bisecting the bleach spot to monitor the recovery, allowing a 4.6 ms frame time using the Zeiss 700. For each bleach acquisition, a background measurement was taken at the same position after waiting 15 s after the bleach acquisition completed. The final spatiotemporal recovery curve was generated by: (1) Normalizing fluorescence and correcting for acquisition bleaching, (2) averaging across the 2-pixel wide line, and (3) finding the bleach-center by Gaussian fitting and then averaging from multiple experiments. First, we fit the mean recovery curve for each condition to both a pure diffusion and reaction-diffusion model (Supplementary Fig. 6)[61–63]. To estimate the experimental error for the parameters, we fit 30 bootstrapped samplings (five experiments per bootstrap) as individual experiments were too noisy to provide accurate fits to the models. The data in Supplementary Table 4 is the mean and standard deviation of each parameter from the 30 bootstraps samples.

For MCF10A, the reaction-diffusion model fit had decreased sum of squared residuals (SSD), while the pure diffusion model provided an unrealistically low diffusion rate, suggesting that significant binding accounts the relatively slow FRAP recovery. For MCF10A+HRas, SUM159, and MDA-MB-231, the reaction-diffusion model slightly decreased SSD compared to pure-diffusion, however the effective diffusion was very large, ranging between 38 and 45 μm²/s, while the pure diffusion model diffusion coefficient ranged between 24 and 26 μm²/s. Previously reported work suggests that the diffusion coefficient for a 100 kDa protein is approximately 21 μm²/s[82]. Therefore, nuclear YAP kinetics in MCF10A+HRas, SUM159, and MDA-MB-231 are best described by pure-diffusion, and binding is negligible at the experimental resolution we could achieve.

**Import/export rates of YAP and synthetic NLS-NES construct**. The first-order nuclear import and export rate of either YAP or the NLS-NES construct (NLS-GFP2X-NES) was measured by monitoring the nuclear intensity after bleaching either the nucleus or cytoplasm. First, a bright field image was used to draw a spline ROI defining the contour of the nucleus or cytoplasm. After bleaching ROI, time-lapse confocal imaging was used to monitor the recovery. We found that full recovery of YAP after nuclear/cytoplasm bleaching was on the time-scale of ~1–2 min. As nuclear volume is essentially constant on this time scale, and YAP signal is homogenous and rapidly diffuses, we measured the average fluorescence intensity of an ROI within the nucleus over time. We then monitored the change of this signal from the first postbleach time point. The first-order import or export rate is then approximated by the slope of a linear fit to the change in signal over the first 30 s (frames 1–4) after bleaching. By dividing the signal trace by the postbleach intensity of the nucleus (export) or cytoplasm (import), we measure the specific first-order transport rate which is independent of protein expression in the cell.

**RNA-seq relative gene expression**. Raw reads of RNA-seq for MCF10A and MCF10AT were downloaded from GEO-Depositions (MCF10A-HRas: GSE81593[83]; MCF10A: GSE75168[84]). The reads were first trimmed with cutadapt

4000

to remove adapters and low-quality reads. The reads then were aligning to human genome hg19 using HISAT2 with default parameters. Gene expression counts were obtained from bam files using htseq-count. The relative expression between MCF10AT and MCF10A was computed per gene as the ratio of counts in MCF10AT divided by counts in MCF10A. This ratio was corrected using the average ratio of 25 housekeeping genes[85–87] between MCF10AT and MCF10A.

**Reporting summary**. Further information on research design is available in the Nature Research Reporting Summary linked to this article.

## Data availability

All oligonucleotide and primer sequences used CRISPR knockin, Genomic PCR, and RT-qPCR can be found in Supplementary Information. Image data files can be made available upon reasonable request. Source data are provided with this paper.

## Code availability

Most of the custom codes used in this article have been uploaded to https://github.com/jmfrank/track_analyzer. Source data are provided with this paper.

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

## Acknowledgements

This work was partially supported by the National Institutes of Health (NIH) National Institute of General Medical Sciences (NIGMS)/National Cancer Institute (NCI) Grant GM77856, NCI Physical Sciences Oncology Center Grant U54CA143836, National Science Foundation Graduate Fellowship Program #DGE-114747, and National Institute of Biomedical Imaging and Bioengineering (NIBIB) 4D Nucleome Roadmap Initiative 1U01EB021237. We thank Lacramioara Bintu (Stanford), Timothy J. Stasevich (Colorado State University) and Pere Roca-Cusachs (Institute of Bioengineering of Catalonia) for their comments on the manuscript. We thank Dr. Barry M. Gumbiner (Center for Developmental Biology and Regenerative Medicine, Seattle Children's Research Institute) for kindly providing us the MCF10A Lats1/2 KO cell line.

## Author contributions

R.P.G., J.M.F., and J.T.L. conceived the project. J.M.F. and R.P.G. designed research. R.P.G. generated the CRISPR/Cas9 genomic knockins. J.M.F. and R.P.G. performed the live imaging experiments. J.M.F. did all FRAP experiments. R.P.G. performed smRNA FISH. Q.S. and R.P.G. performed RT qPCR experiments. R.P.G. and Q.S. performed acrylamide substrate stiffness experiments. R.P.G. and J.M.F. performed IF experiments. M.P.R. generated H1 hESC knock-ins. J.M.F. analyzed the images and generated movies. R.P.G., J.M.F., and J.T.L. wrote the paper.

## Competing interests

The authors declare no competing interests.
