## [Peer Review File · Nature Communications]

Reviewers' comments:

Reviewer #1 (Remarks to the Author):

In this manuscript entitled "Concerted localization resets precede YAP-dependent transcription", the authors developed a cell-based tools for real-time visualization of endogenous YAP in mammary/breast cancer cell lines, and of two YAP target genes (ANKRD1 and AREG), based on CRISPR/CAS9 engineering of the endogenous loci: the endogenous protein (YAP) can be visualized thanks to GFP tagging; the endogenous transcript levels (ANKRD1 and AREG) can be visualized thanks to insertion of multiple MS2 sites in their mRNAs, which can be visualized by expression of a tet-inducible fluorescent MS2-binding protein – thus allowing the very sensitive detection of nascent transcripts within the nucleus as discrete spots.

They then exploit these tools to study YAP localization dynamics and to directly trace nascent YAP-target genes transcripts expression. They validate their system by showing effective exclusion of YAP from the nuclei in MCF10A cells upon dense monolayer growth. This is less efficient in cells expressing the activated HRAS oncogene. They then find that YAP localization in single cells is subjected to fluctuations, lasting hours, and that these fluctuations are reduced (i.e. some cells do not fluctuate anymore) by HRAS and enhanced by the PP1 SRC/RTK inhibitor. Perhaps interesting, these fluctuations occur in a locally coordinated manner in small clumps of cells. This is assumed to be working through LATS kinases, based on the known effects of LATS activation on YAP (i.e. nuclear exclusion).

By examining wound-induced YAP nuclear localization, they realize there are two temporal waves: a very fast one (1-2 min) and a longer one (hours), the latter presumably corresponding to what previously observed in the literature. Given the short times, they make the hypothesis this is related to Calcium fluxes, and show accordingly that thapsigargin or ionomycin (i.e. uncontrolled high cytoplasmic Calcium levels) is sufficient to induce YAP out of the nucleus, albeit with a delayed kinetics (half an our or so – fig 2h), and that this is reversed at least to some extent by inhibition of Calcium-dependent PKC kinase activity.

Surprisingly, they observe that production of nascent transcripts from YAP target genes does not correlate with YAP N/C localization. They also use double thymidine block and find that transcription is initiated shortly after mitosis, when YAP is back on DNA after it has been "excluded" from DNA (i.e. during mitotic chromosome condensation, YAP is no longer on DNA, as many other transcription factors), but after sometimes the frequency of promoter firing is decreased, even if YAP is fully nuclear.

The authors thus hypothesize the existence of a "resetting" system for YAP in normal MCF10A cells: here, YAP is very stable on DNA, and an unknown system would lead to inactivation of this DNA-bound fraction over time; then, to keep YAP transcription at the right level, cells need to "reset" YAP (by a second unknown system) to "refresh" the ppol of YAP on DNA. This system would not be present in transformed cells owing to oncogene activation.

Reviewer comments:

The development of YAP reporter cell lines, and by using advanced techniques to engineer the endogenous loci, is for sure a value of this manuscript. Moreover, the authors focused on cell lines, such as MCF10A and MDAMB231, that are very common cell model systems for the Hippo community, and for the researchers on breast cancer mechanisms.

There are however two main issues to be addressed:

1) to which extent the reporter cell lines do (or do not) confirm what we know about YAP biology. This is important because, presumably, many researchers will ask for these lines and will produce data based on them, assuming GFP-YAP has the same behavior of endogenous YAP just because it was tagged in the endogenous locus; moreover, is itself this technique could be used in future to develop new reporter cell lines /organisms, and a thorough validation is needed to know whether or not the community can invest on those reagents in the future.

2) the authors provide a series of data, very lightly connected to each other, claiming the existence of a "reset" system for YAP activity, which finds little mechanistic dissection, and very little relevance for biology. Moreover, the generality of this should be tested, because the claim it is specific for non-transformed cells is based on the use of one single immortalized cell line.

Specific points:

- the authors should first repeat the most important experiments on YAP regulation to show GFP-YAP and the transcriptional reporters do report correctly on YAP regulation. This may include regulation by mechanical cues, by LATS kinases, by PKA, by GPCRS. Some experiment should also address on parental cell lines (i.e. non-GFP lines) whether endogenous YAP follows the same quantitative / temporal rules than GFP-YAP (obviously, on fixed samples).
- the authors imply multiple times a central role for LATS kinases in the regulation of YAP localization dynamics, but they never prove by an experiment this is the actual mechanism. For example, does knockdown of LATS1/2 rescues YAP N/C resets in confluent and in transformed condition, or in response to oncogene activation? Is this also relevant to regulate YAP target genes and phenotypes?
- the GFP-YAP system would be really useful to understand, based on a semi-endogenous set-up, the kinetics of YAP in different conditions (by FRAP). However, this experiment is limited to comparing different cell lines. The fact that HRAS expression makes YAP shuttling faster both in and out might be interesting, but is not connected with the rest of the story. Moreover, please refrain from using the equation nuclear YAP = DNA-bound or chromatin-bound YAP in FRAP experiments, because this is an assumption and work by others established there is a non-DNA bound fraction in the nuclei (see work from the Sahai group).
- The existence of these “resets” is based on very different experimental systems: confluent cells with YAP excluded from nuclei, confluent cells with YAP excluded from nuclei plus a wound, confluent cells with YAP still in the nucleus, patches of cells, single cells (presumably, in the study of ANKRD1 nascent transcripts). It is hard to understand the experiments, and the rationale behind them, because each and any of these conditions has been already linked to multiple underlying mechanisms, and the authors fail to show what happens in all these systems, what happens to calcium in these different conditions, and then what is the requirement of calcium for YAP in these different conditions. The proposed link with nuclear shape and the nuclear lamina is made in only one single experimental condition, where the existence and relevance of the calcium transients/reset mechanisms have not been demonstrated, and based on a single reference that awaits independent experimental repetition.
- Data on calcium might be in principle interesting, but the authors use a very harsh treatment that will lead to cell death in short time. Is this relevant when calcium is modulated within the physiological ranges, by physiological signals? and then, is this possibly very transient effect on YAP localization and transcription initiation relevant over longer time-scales to drive a phenotype (for example, proliferation etc.)?
- The authors develop interesting tools to directly trace nascent YAP-target genes transcripts expression (ANKRD1 and AREG). However, they found an inverse relation between YAP localization and target gene expression, which remains unexplained. First, the authors do not demonstrate this is also observed upon other more established treatments inducing fast YAP inhibition to the cytoplasm. Then, one question is whether what the authors learn based on these specific target genes / reporters is general for YAP-induced transcription, or just something specific of these promoters. There are techniques to visualize nascent RNA on a whole, and thus to validate what fraction of YAP-dependent transcripts show this behavior upon short treatments.

Reviewer #2 (Remarks to the Author):

The YAP transcriptional coactivator is thought to be primarily regulated by control of its nuclear localization. This manuscript uses single cell real time observation of YAP localization and transcriptional activity to make unexpected discoveries about how YAP activity is regulated in normal and transformed cells that challenge the traditional paradigm. Observations of chromosomally tagged YAP-GFP showed that cells with nuclear YAP undergo fluctuations where YAP will occasionally leave the nucleus and then reaccumulate. Similarly in response to a wound in a confluent monolayer of cells that start with YAP excluded from the nucleus, cells near the wound

rapidly gain nuclear YAP, which is then exported from the nucleus before slowly reaccumulating, a phenomena that involves Ca²⁺ signaling. Rapid YAP export followed by slow reaccumulation in the nucleus can also be triggered by release of intracellular calcium stores. Bursts of transcriptional activity (monitored by observing spots of GFP-MS2 bound to endogenous YAP target RNAs bearing MS2 binding sites) correlate with YAP export followed by import triggered by Ca²⁺ release. Interestingly, HRas transformed cells have reduced rates of fluctuations in YAP nuclear localization yet based on published work have higher YAP activity. The authors show that transformed cells have higher rates of YAP nuclear import and export compared to normal cells. They propose a model where over time YAP in the nucleus becomes inactivated and must be exported from the nucleus to become active so that it can trigger transcription following import into the nucleus.

Overall, this is a quite novel and exciting story that would have broad interest in the Hippo signaling and transcription fields. However, the manuscript has a few issues that should be addressed prior to publication. First, the relationship between transcription burst frequency and YAP activity from population studies should be tested to better show how this work relates to the vast literature describing how YAP activity is regulated by various stimuli (see below). Also because of the brevity of the manuscript there are numerous things that are poorly explained and need to be better described to make the manuscript more understandable.

Major comments.

1) The authors assume that their measurements of transcriptional burst frequency are the same as YAP transcriptional activity as judged in previous experiments looking at mRNA levels of YAP targets in population studies. This is difficult to judge unless they look at transcription burst frequency in individual cells and mRNA levels in populations under the same conditions. This is particularly important because they never look at transcriptional burst frequencies in response to standard manipulations that the field uses to look at regulation of YAP activity such as cell density, F-actin perturbation or wound healing. This is important because in at least one case they see effects contrary to the literature. Specifically, in figure 3 they use the Src inhibitor PP1 to increase LATS and decrease YAP activity (this is problematic for other reasons described below). The authors show that PP1 increases burst frequency, yet published results (for example PMID: 28754671) show a decrease in YAP activity after PP1 treatment. How do the authors account for this? The effect of PP1 treatment on YAP target gene expression (looking at mRNA levels in pools of cells) should be assayed in parallel to see how it compares to their results looking at burst frequencies in single cells.

2) Burst frequencies for YAP target genes should also be analyzed in other situations to determine how relevant they are for YAP regulation. For example, how does cell density (a well characterized regulator of LATS and YAP activity) affect both YAP localization fluctuations and transcriptional burst frequency? In addition the authors should examine transcriptional burst frequency in transformed versus normal cells. In normal unperturbed cells, does burst frequency correlate with nuclear fluctuations of YAP? What about in wound healing assays?

Minor Comments

1) As mentioned above, the authors use activated Ras or a Src inhibitor (PP1) to either decrease or increase LATS activity respectively. If the goal is to understand the contribution of LATS, then these are not good ways to do it since both treatments effect many things besides LATS. Why not knock down LATS to look at the effect of reduction in LATS activity?

2) I have a number of concerns regarding the tagging of endogenous YAP with GFP. For all experiments YAP is tagged at its C-terminus with GFP. This disrupts the PDZ domain binding motif at the C-terminus of YAP. This motif has been shown to be important for nuclear targeting of YAP. The authors show no data testing whether the tag is affecting the function or regulation of YAP. For example is there any effect on YAP target gene expression? Also it is not clear if both alleles of YAP are tagged or if only one, which would make a difference for testing whether YAP function is perturbed.

3) The authors state "Simultaneous treatment of ionomycin and the potent phosphatase inhibitor Okadaic acid, which inhibits protein phosphatase 1 upstream of YAP39, led to a similar degree of cytoplasmic sequestration but with minimal recovery, suggesting that Ca²⁺ drives phosphorylation of YAP followed by PP1-dependent dephosphorylation (Supplementary Figure 2a)." This statement is a large overinterpretation of their data. There is no data presented regarding Ca²⁺ having an effect on YAP phosphorylation under their conditions, and okadaic acid inhibits various phosphatases besides PP1 and could be acting in many different ways. Their model could be stated as one possible interpretation of the data.

4) The authors state that the thapsigargin treatment mimics the effects observed in the wound healing assay. This is not quite true. In the wound healing assay, YAP quickly localizes to the nucleus then leaves, followed by slow accumulation in the nucleus. The thapsigargin treatment causes rapid exit of YAP from the nucleus followed by slow accumulation. The authors should make clear that calcium release just mimics part of the wound healing response. Do the authors have any thoughts about what triggers the initial rapid localization of YAP to the nucleus following wounding?

5) Explanation is sometimes lacking, making it difficult to tell how something was done and what the data is showing. Several examples of this are shown below.

a) The authors state "Immunofluorescence imaging in glioblastoma cells by Liu et al showed that LATS1/2 is critical to Ca²⁺-driven phosphorylation and cytoplasmic sequestration of YAP. Indeed, treatment with the potent protein kinase C inhibitor Go697641 delayed depletion and reduced the magnitude of the localization response (Supplementary Figure 2c, Supplementary Movie 6)." The logic connecting these two sentences is not clear. What the authors are thinking should be spelled out in more detail.

b) What does "neighbors" mean in figure 1D and 1F? This presumably reflects cell density, but it should be spelled out in the legend what it means.

c) Figure 2D measures the "Fraction of cells with at least one fluctuation". The legend says that this measures fluctuations per unit area, but over what time period?

Reviewer #3 (Remarks to the Author):

In this study Franklin et al set out to examine the dynamics and subcellular localization of YAP transcription factor in living cells in conjunction with the transcriptional activity of its target genes, to understand the circuit of activity of YAP dependent gene regulation. The system is based on endogenously tagged YAP proteins as well as the mRNAs transcribed from two chosen target genes (using CRISPR tag knock-ins). This study finds that the nuclear localization of YAP is cell density dependent and that YAP fluctuates between the nucleus and the cytoplasm. These fluctuations can be modulated by interfering with the signal transduction pathway, by administering a wound to the monolayer, and by intracellular calcium release. Intriguingly, the authors noticed fluctuations in nuclear size correlating with the changes in calcium concentration and went on to test whether changing the rigidity of the nucleus with a lamin mutant would influence YAP fluctuations. This was indeed the case.

The authors then followed transcriptional activity of the two target genes using MS2 labeling of the transcripts and infer from the YAP localization data to the transcriptional activity in interphase and after mitosis. It is quite a pity that YAP localization and transcription are not examined in the same cells. Last part of the paper was hard to follow and seems like this was not "milked" to the end... Altogether, the data are of very high quality, novel both technically and scientifically. The experiments performed cover much ground and give a broad picture of the activity and regulation

of this signaling pathway in real time. Basically, this study has the potential for Nature Communications but there are some issues that I have with the manuscript in its current condition:

1) The authors do not do a good job in explaining or interpreting the data and really one has to spend too much effort trying to understand what is it they actually mean to say. They throw a huge amount of data on the page (described succinctly) but don't make much of an effort to go deeper and tell us what it is we learn from this study (seem to skim over the findings), or what are the mechanistics behind the phenotypes.

2) Part of the problem is that the paper is prepared Nature style and this really does not do justice to the study. I would recommend focusing on elaborating, better explanations of the data and figure legends. (In my opinion, less use of elaborate vocabulary and unusual terminology in the Abstract and Introduction would also help the situation). The paper now has 4 main figures. Probably this could be broken down now, panels enlarged, additional supporting information added, all in all trying to make the study more intelligible to the reader.

3) Notably, there is no supplementary molecular or biochemical data (genomic PCR, RT-PCR, Western etc.) to persuade us that the knock-in's worked as expected in all the cell lines, that HRAS is overexpressed in the cells etc. We are expected to take all this for granted but that's not the way things are done. For example, the insertion of both GFP and puromycin – does this affect the final protein (is the scheme drawn to scale)? What about some control experiments with endogenous YAP using antibodies to show that one sees similar phenotypes of the endogenous and fusion proteins, or RNA FISH for the transcription part.

Other points:

Abstract – “concerted localization resets” – the reader has no way of understanding this peculiar phrase when reading the abstract. In fact the term “localization reset” is never explained. The whole sentence is even more confusing – “concerted localization resets, which are dramatic, concerted departure/reentry cycles”

Fig. 1C – unclear what is “counts” in the y-axis and it takes time realize that the densities on the right plots refer also to the left hand plots (legend does not give much explanation).

Fig. 1D – “neighbors” in axis - supposedly you mean no. of neighboring cells? Is it possible that a cells has 10's of neighbors?

Movie 1- the density in the second part of the movie does not reach the same levels as in the first part. As for cell density – there could have been more analysis of this issue.

Fig. 2B – no numbers on y axis

Fig. 2C – no explanation. What do the plots tell us? How can there be a 0 for N/C?

Fig. 2E – how frequent is this phenomenon?

Fig. 2G – only one row of cells was influenced? If so, this should be mentioned.

Fig. 2H – are these frames from a time lapse movie? Are the times in 2J the same for all? Is the first image before or after treatment? Line 118 – “resulting in a bulk YAP translocation cycle similar to that seen at the wound edge” – actually it's not the same. Importantly, it is a pity that the calcium and YAP are not imaged together in the same cell (and to show this also for the wounding experiment).

Line 97 – “The fluctuation magnitude and frequency were unchanged across conditions” – this remains as a statement with no elaboration or conclusion and these parameters are not introduced in any way. Actually, PP1 shows some difference.

Line 99 – PP1 – no explanation what this is

Movie 3 – don't really see any reduction in N/C in control cells. Maybe add arrows to point to the cells since its moving so fast.

Line 103 – “fast localization fluctuations of YAP” – how fast?

Fig. 3 – Here too no controls. For instance, a smFISH experiment to show how the gene acts under regular conditions prior to knock-ins and that it doesn't change after the knock-in. Importantly, as mentioned above, show the RNA and protein in the same cell. If this is hard to do in live cells, then at least in fixed cells it can be shown (Ab to YAP in gene-MS2 cells, and smFISH in the YAP-eGFP cells). What about a control showing an unaffected gene under same conditions and treatments (doesn't have to be MS2 tagged).

Supplementary Figure 4 – no real explanations or interpretation/discussion

Figure 4 –nothing here is well explained. Lots of data condensed together – really one cannot easily follow or try to review this part (not to mention the non-regular presentation of the FRAP data that needs explaining) and understand what is the biological significance of this information. Discussion – “chromatin interactions” - not really something measured in this study
Number the movie files.

We thank the reviewers for their positive comments, and we are happy to see their overall endorsement of our work.

The revised manuscript is now ~5300 words long, has **8** main text figures (**new main text Figure panels** are Figures 1b-1h, 2f-g, 3e-h, 4e-f, 5b-i and 6g-m), **2** new Supplementary Figures (**new Supplementary Figure panels** are Figures 1a-1h, 3g-h, 4a-4b) and **5** new Supplementary Movies (movies 1, 5, 6, 10, 11) which represents multiple new experiments and also a redistribution of earlier figure components to help ease the flow of the paper. The new text, figures and the supplementary figures include:

1. A detailed discussion of the contexts in which potential long term effects of YAP fluctuations/localization-resets may be relevant and the approach that would be essential to track long term effects of transient fluctuations.
2. Considerably more detailed explanations of experimental design, findings, and conclusions.
3. YAP-eGFP knockin validation by genomic PCR and capillary western blot.
4. Validation of YAP-eGFP localization through co-staining of native YAP using immunofluorescence.
5. Assessment of the performance of MCF10A^{YAP-GFP-KI} cell line in substrate stiffness sensing, by monitoring YAP-eGFP localization on acrylamide hydrogels of a range of stiffnesses.
6. Assessment of changes in YAP-eGFP localization in response to changes in actomyosin contractility.
7. Assessment of changes in YAP-eGFP localization in response to serum starvation followed by serum replenishment or Lysophosphatidic acid (LPA) treatment.
8. Immunostaining based analysis of localization-reset time course in wild-type and Lats1/2 knockout MCF10A cell line to definitively show that calcium induced localization-reset of YAP is directly dependent on LATS kinase activity.
9. YAP fluctuation dynamics and density dependent changes in localization in YAP-eGFP knock-in H1 hESC embryonic stem cell line.
10. MS2-transcription reporter cassette knock-in validation using genomic PCR and single molecule RNA FISH.
11. Measurement of nascent transcription dynamics in MCF10A^{AREG-MS2-KI} and in MCF10A^{Ankrd1-MS2-KI} cells at different monolayer densities, to compare the performance of dynamic measurement of nascent transcription to known estimates of transcription based on more conventional measurements.
12. Measurement of nascent transcription dynamics in serum-starved MCF10A^{AREG-MS2-KI} and in MCF10A^{Ankrd1-MS2-KI} cells upon serum replenishment, to compare the performance of dynamic measurement of nascent transcription to known estimates of transcription based on more conventional bulk measurements.
13. Single molecule RNA FISH measurements of the effect of calcium induced YAP localization-reset on AREG (YAP target gene) and GAPDH (housekeeping gene) expression.
14. RT qPCR analysis of multiple YAP target-gene expression (AREG, Ankrd1, CTGF, IGFBP3, TGFB2, Foxf2, RASSF2, Cry2) and housekeeping gene expression (GAPDH, HPRT1, PGK1) using pooled mRNA from populations of cells during different time points of a Ca²⁺ induced localization-reset cycle.
15. Assessment of the effect of Lamin Del50 on the kinetics and amplitude of SOCE induced by Thapsigargin treatment.

New experiments, methods and figure legends added to the main text in response to reviewers' comments have been highlighted **yellow**.

Detailed clarification of earlier data, and new discussion sections added the main text in response to the reviewers' comments have been highlighted **blue**.

Please see below for a detailed point by point response to all comments.

Reviewer #1 (Remarks to the Author)

In this manuscript entitled “Concerted localization resets precede YAP-dependent transcription”, the authors developed a cell-based tools for real-time visualization of endogenous YAP in mammary/breast cancer cell lines, and of two YAP target genes (ANKRD1 and AREG), based on CRISPR/CAS9 engineering of the endogenous loci: the endogenous protein (YAP) can be visualized thanks to GFP tagging; the endogenous transcript levels (ANKRD1 and AREG) can be visualized thanks to insertion of multiple MS2 sites in their mRNAs, which can be visualized by expression of a Tet-inducible fluorescent MS2-binding protein – thus allowing the very sensitive detection of nascent transcripts within the nucleus as discrete spots.

They then exploit these tools to study YAP localization dynamics and to directly trace nascent YAP-target genes transcripts expression. They validate their system by showing effective exclusion of YAP from the nuclei in MCF10A cells upon dense monolayer growth. This is less efficient in cells expressing the activated HRAS oncogene. They then find that YAP localization in single cells is subjected to fluctuations, lasting hours, and that these fluctuations are reduced (i.e. some cells do not fluctuate anymore) by HRAS and enhanced by the PP1 SRC/RTK inhibitor. Perhaps interesting, these fluctuations occur in a locally coordinated manner in small clumps of cells. This is assumed to be working through LATS kinases, based on the known effects of LATS activation on YAP (i.e. nuclear exclusion).

By examining wound-induced YAP nuclear localization, they realize there are two temporal waves: a very fast one (1-2 min) and a longer one (hours), the latter presumably corresponding to what previously observed in the literature. Given the short times, they make the hypothesis this is related to Calcium fluxes, and show accordingly that Thapsigargin or Ionomycin (i.e. uncontrolled high cytoplasmic Calcium levels) is sufficient to induce YAP out of the nucleus, albeit with a delayed kinetics (half an our or so – fig 2h), and that this is reversed at least to some extent by inhibition of Calcium-dependent PKC kinase activity.

Surprisingly, they observe that production of nascent transcripts from YAP target genes does not correlate with YAP N/C localization. They also use double thymidine block and find that transcription is initiated shortly after mitosis, when YAP is back on DNA after it has been “excluded” from DNA (i.e. during mitotic chromosome condensation, YAP is no longer on DNA, as many other transcription factors), but after sometimes the frequency of promoter firing is decreased, even if YAP is fully nuclear.

The authors thus hypothesize the existence of a “resetting” system for YAP in normal MCF10A cells: here, YAP is very stable on DNA, and an unknown system would lead to inactivation of this DNA-bound fraction over time; then, to keep YAP transcription at the right level, cells need to “reset” YAP (by a second unknown system) to “refresh” the pool of YAP on DNA. This system would not be present in transformed cells owing to oncogene activation.

General comments: Reviewer 1

The development of YAP reporter cell lines, and by using advanced techniques to engineer the endogenous loci, is for sure a value of this manuscript. Moreover, the authors focused on cell lines, such as MCF10A and MDAMB231, that are very common cell model systems for the Hippo community, and for the researchers on breast cancer mechanisms.

Response. We thank the reviewer for recognizing the value of our work and suggesting insightful control experiments to further validate our findings.

There are however two main issues to be addressed:

General comment 1. To which extent the reporter cell lines do (or do not) confirm what we know about YAP biology. This is important because, presumably, many researchers will ask for these lines and will produce data based on them, assuming GFP-YAP has the same behavior of endogenous YAP just because it was tagged in the endogenous locus; moreover, is itself this technique could be used in future to develop new reporter cell lines /organisms, and a thorough validation is needed to know whether or not the community can invest on those reagents in the future.

Response. In our initial submission, we had shown that YAP-eGFP knockin lines exhibit density dependent cytoplasmic sequestration which is well known phenotype of YAP distribution in sparse vs. dense cells based on earlier immunofluorescence data.

Per reviewer's suggestion, we have now carried out a wide range of tests to evaluate the effects of perturbations with known outcomes on YAP localization in the MCF10A^{YAP-GFP-KI} cell line. Firstly, we now provide the YAP-eGFP knockin validation by genomic PCR and capillary western blot (**Figure 1b, Supplementary figure 1**). We have also validated YAP-eGFP localization through co-staining of native YAP using immunofluorescence (**Figure 1c-d**). Further, we have now assessed the performance of the MCF10A^{YAP-GFP-KI} cell line in substrate stiffness sensing

(**Figure 1e-f**), response to Latrunculin B treatment (**Figure 1g-h**), and serum starvation and replenishment or Lysophosphatidic acid (LPA) stimulation (**Figure 3e-f**).

Not only did these experiments reproduce previously reported effects based on biochemical and immunofluorescence assays, but by adding a temporal dimension, we were able to provide additional details about the dynamics of the responses. In the case of serum starvation followed by serum replenishment, time-lapse imaging revealed yet another context where YAP localization showed dynamic fluctuations in response to an external signaling cue (**Figure 3e-f**).

General comment 2a. The authors provide a series of data, very lightly connected to each other, claiming the existence of a “reset” system for YAP activity, which finds little mechanistic dissection, and very little relevance for biology.

Response. The main contention in our manuscript is that YAP nucleocytoplasmic shuttling, and not the mere presence of nuclear YAP, is key in regulating the transcriptional output of YAP. Cells at steady-state have constant turnover of YAP between the nucleus and cytoplasm. Therefore, the reset of YAP can occur through the total redistribution of most YAP molecules from the nucleus to cytoplasm and back (**explored in Figures 3-7**) **OR** through the baseline stochastic exchange of individual molecules across the nuclear membrane (**explored in Figure 8**).

In our initial submission, we had pointed out that coordinated localization-reset of YAP over many adjacent cells is a rare event, which makes it challenging to rely purely on spontaneous fluctuations to test the impact of localization reset on transcription. We discovered that calcium stimulation could drive YAP localization-reset uniformly across all cells, allowing us to robustly assess the transcriptional response across the sample in a kinetically controlled manner.

Additionally, few recent works have demonstrated YAP nucleocytoplasmic fluctuations in mammalian cells, albeit using exogenous overexpression of YAP-FP fusions, and in *Drosophila* cells expressing YAP-GFP generated through genomic knockin. Please see introduction section for a brief discussion of these earlier works.

A recent study showed that treatment with Angiotensin II causes a similar localization-reset of YAP. Further, transcription activation of YAP target genes by Angiotensin II was contingent on the ability of YAP to shuttle (Wang et al. *J. Biol. Chem.*, 2016). While imaging of fixed immunofluorescence samples used in this study presented only static snapshots of YAP localization, it nonetheless supports the possibility that localization-resets may be fundamental to a wide variety of signaling pathways and cell lines. Given that such localization-resets are observed under a variety of signaling contexts in multiple cell lines, primary cells, and in vivo, we believe these observations to be biologically relevant. Importantly, this mode of regulation by localization-reset has also been shown to govern the transcriptional activity of another transcription factor, Nrf2, as already referenced in our manuscript.

For further mechanistic dissection of localization-reset, we have now used a LATS1/2 double-knockout cell line to definitively show that calcium-induced localization-reset of YAP is dependent on LATS kinase activity (**Figure 4e**).

Although resets are relatively rare, our results indicate that the baseline transport rates across the nuclear membrane may act as a transcription gate. We show that mere activation of Ras abolishes localization-resets and alters YAP-chromatin binding to allow rapid baseline nucleocytoplasmic exchange. Multiple transformed cell lines also show similarly altered transport, which may contribute to oncogenesis.

General Comment 2b. Moreover, the generality of this should be tested, because the claim it is specific for non-transformed cells is based on the use of one single immortalized cell line.

Response. To extend our work beyond a single immortalized cell line, we generated YAP-eGFP genomic knockin of H1 human embryonic stem cell line (**H1-hESC**). This cell line showed YAP cytoplasmic sequestration at very high densities (**Figure 2f-g**). Using time-lapse microscopy, we observed dynamic fluctuations in YAP N/C ratios in H1-hESCs at similar timescales to MCF10A breast epithelial cells, demonstrating that fluctuations in YAP localization is not limited to somatically differentiated cells (**Figure 3i-j**).

Moreover, Ege et al. 2018 has shown similar short time scale fluctuations in N/C ratios albeit using lentiviral overexpression of YAP-eYFP in normal mammary fibroblasts. We had referenced this paper in our earlier submission. We have now provided more detail in the introduction section. Here's the relevant excerpt from the main text.

*“A recent study of Yorkie (*Drosophila* YAP homolog) subcellular localization and dynamics in *D. melanogaster*¹⁴ and several studies using live imaging in mammalian cells expressing exogenous YAP^{15,16} have reported rapid shuttling between cytoplasm and nucleus.”*

Specific comments: Reviewer 1

Specific comment 1. The authors should first repeat the most important experiments on YAP regulation to show GFP-YAP and the transcriptional reporters do report correctly on YAP regulation. This may include regulation by mechanical cues, by LATS kinases, by PKA, by GPCRS.

Response. We thank the reviewer for suggesting these validation experiments. As mentioned in response to **General comment 1** (see above), we have now validated the function of YAP-eGFP in great detail. To test the robustness of the 3' UTR-(MS2 transcriptional cassette)-knockin cell lines in reporting known effects of YAP compartmentalization on transcriptional regulation based on earlier ensemble measurements, we have measured the nascent transcriptional response in two well-characterized biological contexts: density sensing and serum stimulation (**Figures 5c-i**).

First, we looked at baseline transcriptional frequency and amplitude in sub-confluent (nuclear YAP), and over-confluent (cytoplasmic YAP) cultures. Consistent with a large body of literature, our transcriptional reporter lines showed significantly lower activity at high density, measured in terms of nascent transcriptional spot frequency (**Figure 5d-e**).

Second, we assessed the dynamic transcriptional response to serum replenishment after 20 hours of serum starvation. Lysophosphatidic acid (LPA) present in serum has been shown to activate GPCRs, thereby inhibiting the Hippo pathway and causing nuclear relocation of transcriptionally active YAP (Yu et al. Cell. 2012). Per literature, we found very few cells actively transcribing after serum starvation. Upon addition of serum, we observed a transient burst in transcriptional activity in the monolayer culture (**Figure 5f-g**), in line with previously published steady-state data (Yu et al. Cell. 2012). The observation that serum replenishment only transiently upregulated YAP dependent transcription provides further support to our contention that transcriptional activity is not just a simple function of the nuclear abundance of YAP, as YAP is mostly nuclear following serum stimulation (**Figure 3e**).

Specific comment 2. Some experiment should also address on parental cell lines (i.e. non-GFP lines) whether endogenous YAP follows the same quantitative / temporal rules than GFP-YAP (obviously, on fixed samples).

Response. To address this point, we monitored calcium-induced YAP localization-reset through immuno-staining of native YAP in MCF10A cell line after Thapsigargin treatment, every 20 minutes for 140 minutes, essentially providing relatively high-frequency montage of the YAP localization-reset timeline. These measurements of native YAP localization reset in parental MCF10A cell-line were consistent with the YAP-eGFP knock-in cell line (**Figure 4e-f**).

Specific comment 3. The authors imply multiple times a central role for LATS kinases in the regulation of YAP localization dynamics, but they never prove by an experiment this is the actual mechanism.

Response. In our earlier submission, we had shown that the small molecule inhibitor Go6976, which inhibits cPKCs, only partly dampened calcium-induced YAP localization-reset (Liu et al. Oncogene 2019). To test whether LATS kinases played a direct role in calcium-induced YAP localization reset, we obtained an MCF10A LATS1/2 knock-out cell line from the Gumbiner lab (Kim and Gumbiner PNAS 2019) and measured YAP N/C ratios at varying time points after Thapsigargin treatment using immunofluorescence staining. Our data clearly shows that LATS1/2 knock-out completely abolished YAP nuclear depletion seen upon treatment with Thapsigargin in native MCF10A cells (**Figure 4e-f**).

Specific comment 4. The GFP-YAP system would be really useful to understand, based on a semi-endogenous set-up, the kinetics of YAP in different conditions (by FRAP). However, this experiment is limited to comparing different cell lines. The fact that HRAS expression makes YAP shuttling faster both in and out might be interesting, but is not connected with the rest of the story.

Response. In our initial submission, to observe brevity, we had not elaborated on the connections between each aspect of the story sufficiently. We have now clearly stated the rationale connecting the different facets of our work.

Our findings show that cellular transformation in multiple cell lines (10AT, SUM159, MDA-MD-231) dramatically increases YAP nucleocytoplasmic shuttling rates (**Figure 8c**) while dampening spontaneous localization reset probability (**Figure 3d**). We have also shown that YAP shuttling dynamics is strongly correlated with downstream transcription control (**Supplementary Figure 6**).

Additionally, oncogenic transformations have been shown to cause sustained YAP activation in a wide range of scenarios, up-regulating transcription of YAP target genes and promoting tumor development.

Our data suggest that markedly higher nucleocytoplasmic exchange rates in transformed cells may maintain YAP in a hyperactive state. Essentially, while spontaneous localization-resets may occasionally reboot YAP activity in normal cells, oncogenically transformed cells have escaped the need for such reset by maintaining a much higher nucleocytoplasmic shuttling rate of YAP. Such a kinetic model may be transformative towards our understanding of the basis of YAP transcriptional deregulation in cancer. Mechanistic dissection in our future work will help identify molecular players and possibly new targets for inhibiting YAP deregulation.

Specific Comment 5. Moreover, please refrain from using the equation nuclear YAP = DNA-bound or chromatin-bound YAP in FRAP experiments, because this is an assumption and work by others established there is a non-DNA bound fraction in the nuclei (see work from the Sahai group).

Response. The reviewer is correct in suggesting not to equate nuclear bound YAP to DNA-bound YAP. Indeed, we did not equate nuclear bound YAP to DNA-bound YAP in our original manuscript. We refer the reviewer to our original FRAP data (now **current Figure 8d, Supplementary Figure 7, and Supplementary Table 1**), where we had used highly resolved spatiotemporal FRAP measurements of the nuclear pool of YAP. By fitting the data to pure diffusion and reaction-diffusion models, we precisely calculated the bound fraction of YAP in normal and transformed cells.

The reason for carrying out spatiotemporal FRAP was our hypothesis that higher levels of YAP in the nucleus (as is the case with Ras-transformed cells) do not necessarily mean a larger fraction of bound YAP. Please see **Figure 8d and Methods** for further details. We have further clarified our results and the need for carrying out spatiotemporal FRAP in the current manuscript.

Specific Comment 6. The existence of these “resets” is based on very different experimental systems: confluent cells with YAP excluded from nuclei, confluent cells with YAP excluded from nuclei plus a wound, confluent cells with YAP still in the nucleus, patches of cells, single cells (presumably, in the study of ANKRD1 nascent transcripts). It is hard to understand the experiments, and the rationale behind them, because each and any of these conditions has been already linked to multiple underlying mechanisms, and the authors fail to show what happens in all these systems, what happens to calcium in these different conditions, and then what is the requirement of calcium for YAP in these different conditions.

Response. We believe the observations of YAP localization-resets in many different experimental contexts lend support to our claims that this mechanism may be *universal rather than specific*. We now add that resets are seen in human embryonic stem cells and cells responding to serum/LPA, further extending the examples of resets in different contexts.

Although we believe the exploration of the mechanism in all these contexts is beyond the scope of a single paper, we now provide more mechanistic detail for the Ca^{2+} -induced reset scenario, clearly establishing through the use of Lats1/2 knockout cells, that Ca^{2+} -induced localization-reset is directly dependent on LATS kinases (**Figure 4e, f**)

Specific Comment 7. The proposed link with nuclear shape and the nuclear lamina is made in only one single experimental condition, where the existence and relevance of the calcium transients/reset mechanisms have not been demonstrated, and based on a single reference that awaits independent experimental repetition.

Response. In our initial submission, we had pointed out that SOCE upon Thapsigargin treatment lead to marked compression of the nuclear membrane. Since actomyosin contractility and nuclear mechanics have been directly connected with YAP localization, we wanted to know whether changes in nuclear mechanics could at least in part be responsible for the observed YAP localization reset. The reason we chose Lamin del50 as our perturbation, was to alter the mechanical properties of the nuclear membrane in a controlled manner without affecting actomyosin contractility itself and assess YAP localization changes in response to calcium release. This system is inducible, allowing us to transiently perturb cells through lamin-del50 expression, minimizing long-term effects. We find that cytoplasmic translocation of YAP was reduced after calcium release in lamin-del50 expressing cells, suggesting a potential connection between nuclear mechanics and YAP localization-reset (**Supplementary Figure 3d-f**). **We have now included a plot of Ca^{2+} transient in cells expressing Lamin del50 showing that SOCE itself is not affected by Thapsigargin treatment (Supplementary Figure 3g-h).**

Specific Comment 8. Data on calcium might be in principle interesting, but the authors use a very harsh treatment that will lead to cell death in short time. Is this relevant when calcium is modulated within the physiological ranges, by physiological signals?

Response. Our initial interest in calcium signaling was due to the fast scale YAP localization reset in cells at the edge of an epithelial wound (**Figure 4a**). In this context, calcium signaling has been thoroughly explored as a biologically relevant signal to drive the wound healing response. In order to explore calcium signaling in a controlled and statistically robust manner, researchers frequently use Thapsigargin or Ionomycin to induce intracellular calcium release (Liu et al. *Oncogene* 2019; Wales et al. *eLife* 2016). We have used this probe to provide a distinct and robust signal to the cell and observe the effect(s) on YAP. Indeed, at very long time-scales, Thapsigargin treatment can lead to cell death, but our experimental timeline is confined to 2hrs for imaging and 3 hours for RT qPCR. Additionally,

over our experimental duration, we have not observed any change expression of 3 different housekeeping genes (**Figure 6m**).

Specific comment 9. and then, is this possibly very transient effect on YAP localization and transcription initiation relevant over longer time-scales to drive a phenotype (for example, proliferation etc.)?

Response. We thank the reviewer for pointing this out. Isolating the long-term effects of YAP localization resets independently from other signaling inputs would require precise external control of native YAP localization dynamics, which is beyond the scope of this current work. However, this a highly relevant future direction. We have discussed this in detail in the manuscript.

“While our results demonstrate a strong correlation between YAP localization-reset and gene expression, an important next step would be to develop a live imaging platform to establish the temporal relationship between YAP localization fluctuation and transcriptional activation at single-cell resolution. Another future challenge will be to assess how differences in magnitude and frequency of fluctuations in otherwise phenotypically identical cells could potentially affect cellular fate. Fluctuations in gene expression are known to alter cell-fate decisions⁶⁴, and YAP has been implicated in stem cell pluripotency.⁶⁵ Transient reversals in YAP localization acting as a fast trigger of YAP activity could be relevant for spontaneous cell fate switching in stem cells as well as lineage reprogramming.^{66,67} Combining real-time detection of dynamic fluctuations in YAP localization with single-cell genomics^{68,69} and lineage tracing^{70,71} will help identify long term effects of YAP fluctuation on cell fate decisions. “

Specific comment 10. The authors develop interesting tools to directly trace nascent YAP-target genes transcripts expression (ANKRD1 and AREG). However, they found an inverse relation between YAP localization and target gene expression, which remains unexplained. First, the authors do not demonstrate this is also observed upon other more established treatments inducing fast YAP inhibition to the cytoplasm. Then, one question is whether what the authors learn based on these specific target genes / reporters is general for YAP-induced transcription, or just something specific of these promoters. There are techniques to visualize nascent RNA on a whole, and thus to validate what fraction of YAP-dependent transcripts show this behavior upon short treatments.

Response. While aiming for brevity in our initial submission we did not present several significant points with enough clarity. We do *not* propose that YAP nuclear localization and transcription are inversely correlated. We propose that a rapid nucleocytoplasmic reset (nuclear depletion followed by nuclear replenishment cycle) induces transient activation of YAP target genes. we have now clarified this point. Here’s a relevant excerpt from the main text

“Next, we asked whether YAP localization-resets (nuclear depletion followed by re-enrichment) affected the expression of YAP target genes.”

In this context, we would need assays where YAP is rapidly exported and then imported (calcium, cell division: **Figures 4 and 7**), or YAP is initially cytoplasmic and then rapidly imported (serum starvation followed by replenishment or LPA treatment: **Figure 3e**). In our current submission, we have included new experiments where we show that serum starvation followed by replenishment causes a transient spike in AREG and Ankrd1 expression, which decays upon prolonged nuclear retention (**Figure 5**).

As explained earlier, since our hypothesis is not that nuclear localization itself is sufficient for YAP activation, perturbations to solely drive YAP to the cytoplasm do not provide means to test our central hypothesis.

The reviewer comment brings up an important point – for our transcription assays, to what extent are the effects specific to YAP driven transcription, and then, to what extent are known YAP target genes activated. First, we performed single-molecule RNA FISH against the housekeeping gene GAPDH after Thapsigargin treatment and found no significant change in cell frequency, and a modest change in transcription amplitude, as compared to the substantial changes in AREG transcription (**Figure 6g-l**). We further explored the specificity of the response using RT-qPCR analysis of multiple YAP target-gene expression (AREG, Ankrd1, CTGF, IGFBP3, TGFB2, Foxf2, RASSF2, CYR61) and housekeeping gene expression (GAPDH, HPRT1, PGK1) through the course of induced YAP localization reset (**Figure 6m**). In agreement with our nascent transcription data, we see general activation of YAP target genes, with minimal activation of house-keeping genes (**Figure 6m**).

We agree with the reviewer that techniques do exist to visualize nascent RNA on the whole, albeit at much lower temporal resolution, such as GRO-seq, PRO-seq (Genomics approaches), or MERFISH (single molecule FISH approaches). While establishing these protocols for the range of nascent transcription kinetics experiments we have done for this paper is beyond the scope for current work, it is a worthy future direction. Because our new RT qPCR data for the Ca²⁺ release assay shows the response for an array of YAP targets consistent with earlier work, with no significant response for multiple housekeeping genes, we believe this provides ample evidence to show general YAP

activation. Further, we do not necessarily expect that every known YAP target gene will be activated under each stimulus.

Reviewer #2 (Remarks to the Author):

The YAP transcriptional coactivator is thought to be primarily regulated by control of its nuclear localization. This manuscript uses single cell real time observation of YAP localization and transcriptional activity to make unexpected discoveries about how YAP activity is regulated in normal and transformed cells that challenge the traditional paradigm. Observations of chromosomally tagged YAP-GFP showed that cells with nuclear YAP undergo fluctuations where YAP will occasionally leave the nucleus and then reaccumulate. Similarly, in response to a wound in a confluent monolayer of cells that start with YAP excluded from the nucleus, cells near the wound rapidly gain nuclear YAP, which is then exported from the nucleus before slowly reaccumulating, a phenomenon that involves Ca²⁺ signaling. Rapid YAP export followed by slow reaccumulation in the nucleus can also be triggered by release of intracellular calcium stores. Bursts of transcriptional activity (monitored by observing spots of GFP-MS2 bound to endogenous YAP target RNAs bearing MS2 binding sites) correlate with YAP export followed by import triggered by Ca²⁺ release. Interestingly, HRas transformed cells have reduced rates of fluctuations in YAP nuclear localization yet based on published work have higher YAP activity. The authors show that transformed cells have higher rates of YAP nuclear import and export compared to normal cells. They propose a model where over time YAP in the nucleus becomes inactivated and must be exported from the nucleus to become active so that it can trigger transcription following import into the nucleus.

Overall, this is a quite novel and exciting story that would have broad interest in the Hippo signaling and transcription fields. However, the manuscript has a few issues that should be addressed prior to publication. First, the relationship between transcription burst frequency and YAP activity from population studies should be tested to better show how this work relates to the vast literature describing how YAP activity is regulated by various stimuli (see below). Also because of the brevity of the manuscript there are numerous things that are poorly explained and need to be better described to make the manuscript more understandable.

We thank the reviewer for recognizing that this is a “novel and exciting story that would have broad interest in the Hippo signaling and transcription fields.”

Major comments: Reviewer 2

Major comment 1. The authors assume that their measurements of transcriptional burst frequency are the same as YAP transcriptional activity as judged in previous experiments looking at mRNA levels of YAP targets in population studies. This is difficult to judge unless they look at transcription burst frequency in individual cells and mRNA levels in populations under the same conditions. This is particularly important because they never look at transcriptional burst frequencies in response to standard manipulations that the field uses to look at regulation of YAP activity such as cell density, F-actin perturbation or wound healing. This is important because in at least one case they see effects contrary to the literature.

Response. We thank the reviewer for pointing out the need for looking at nascent transcription metrics in the context of well defined YAP-dependent transcription assays. To address this concern, we analyzed nascent transcription burst frequency and amplitude in AREG and Ankrd1 transcriptional reporter cell lines grown to different degrees of confluence (sparse vs. dense) (**Figure 5d-e**). We also carried out nascent transcription analysis of AREG and Ankrd1 transcriptional reporter cell lines upon serum starvation and subsequent serum replenishment (**Figure 5f-g**). We found that transcriptional burst frequency in our nascent transcription reporter lines recapitulated the previously established effects of cell density and serum-(starvation-replenishment) cycle on YAP signaling.

Additionally, to look at mRNA levels in populations of cells under identical experimental conditions, as suggested by the reviewer, we pooled mRNA from populations of cells during different time points of a Ca²⁺ induced localization-reset cycle. Using this pooled mRNA, we carried out RT qPCR analysis of multiple YAP target-gene expression (AREG, Ankrd1, CTGF, IGFBP3, TGFB2, Foxf2, RASSF2, CYR61) and housekeeping gene expression (GAPDH, HPRT1, PGK1). The qPCR results show a general agreement with our dynamic nascent transcription analysis (**Figure 6m**), showing activation of YAP target genes, with minimal activation of house-keeping genes (**Figure 6m**). These results give us confidence in using transcriptional burst measurements of native YAP target genes to assess YAP activity.

Major comment 2. Specifically, in figure 3 they use the Src inhibitor PP1 to increase LATS and decrease YAP activity (this is problematic for other reasons described below). The authors show that PP1 increases burst frequency, yet published results (for example PMID: 28754671) show a decrease in YAP activity after PP1 treatment. How do the authors account for this? The effect of PP1 treatment on YAP target gene expression (looking at mRNA levels in

pools of cells) should be assayed in parallel to see how it compares to their results looking at burst frequencies in single cells.

We thank the reviewer for raising an interesting point regarding the effect of Src inhibition on YAP activity. We want to point out that current literature often uses additional constraints to assay the effect of Src inhibition. For example, in the referred paper (pubmedID:28754671), the authors used a Trypsinization followed by adhesion cycle to test the effects of Src inhibition by dasatinib. Another work used prolonged serum starvation, followed by Src inhibition to show YAP is cytoplasmic. These assays utilize considerably different base signaling environments than our's since both mechanical alteration of cytoskeletal properties (Trypsinization followed by adhesion) and prolonged serum starvation on their own would have marked impact on YAP's localization and activity. In our work, instead of looking at the cumulative impact of Src inhibition and other forms of signaling manipulation, we look at Src inhibition in sub-confluent, unperturbed cultures without imposing additional constraints. We have now clarified this in the manuscript in sufficient detail.

Here's the relevant excerpt from the main text.

"Previous reports have shown uniform and permanent changes in YAP localization after Src inhibition in serum-starved cells³⁴, and when used in conjunction with Trypsinization-reattachment cycle.³⁵ To decouple the effect of Src kinase inhibition from other forms of modulations such as serum starvation³⁶ or cell detachment/adhesion cycles³⁷ we treated sub-confluent MCF10A cells grown in complete media with PP1, a potent inhibitor of Src kinase.³⁸ Surprisingly, we found that treatment with 1 μ M PP1 in complete growth media lead to exaggerated localization fluctuations rather than uniform cytoplasmic sequestration (Figure 3c, Supplementary Movie 4)."

As for comparing mRNA levels in pools of cells, please see response to comment 1a.

Major comment 2. Burst frequencies for YAP target genes should also be analyzed in other situations to determine how relevant they are for YAP regulation. For example, how does cell density (a well characterized regulator of LATS and YAP activity) affect both YAP localization fluctuations and transcriptional burst frequency? In addition, the authors should examine transcriptional burst frequency in transformed versus normal cells.

Response. As discussed above under response to comment 1a, we analyzed transcriptional burst frequency in AREG and Ankrd1-MS2 knockin lines at varying levels of cell density (**Figure 5d-e**) which is an essential modulator of LATS kinase activity, as well as serum starvation-replenishment cycle (**Figure 5f-g**), which is an important modulator GPCR signaling. These experiments establish that analysis of nascent transcription kinetics recapitulates results derived from ensemble experiments while providing a dynamic perspective of the response.

While we agree with the reviewer that the analysis of transcriptional burst frequency in transformed cells will be interesting direction, we prioritized other experiments over generating a triple stable (3 generation of selection) MCF10A cell line required for imaging transcription kinetics in HRas transformed MCF10A cell line (MS2 knockin using CRISPR + MCP-mNeon expression using Piggy-BAC transposition + HRas expression using lentiviral transduction). In our experience, MCF10A is quite sensitive to genetic perturbations, antibiotic selections, and cell culture passages and beyond two genetic alterations, the cells seem to transform morphologically. Importantly, by analyzing existing RNA-seq data in the literature, we have shown that Ras transformation of MCF10A upregulates many YAP target genes (**Supplementary Figure 6**).

Major comment 3. In normal unperturbed cells, does burst frequency correlate with nuclear fluctuations of YAP? What about in wound healing assays?

As explained above, following burst frequency in cells showing spontaneous fluctuations would require a triple stable cell line (MS2 knockin using CRISPR + MCP-Halo expression using Piggy-BAC transposition + YAP-GFP knockin using CRISPR) which is inherently difficult with MCF10A cells, since they start showing signs of phenotypic transformation with increasing cell culture passages. In this particular setup we would not even be able to minimize the number of steps involved in generating the triple stable cell lines through the use of polycistronic vectors, since two of the steps involve generating unique CRISPR knockins.

To address this specific question, we would need to generate dual YAP knockin and target gene-MS2 knockins of cell lines that are more resistant to cell passage related transformations, validate the lines and then carry out fluctuation versus transcription kinetics in real time in two color. While this is a very relevant direction and worth pursuing as a next step, it is beyond the scope and timeline of the current work. We have added this as an important future direction in the discussion section.

Here's the relevant excerpt from the main text.

“While our results demonstrate a strong correlation between YAP localization-reset and gene expression, an important next step would be to develop a live imaging platform to establish the temporal relationship between YAP localization fluctuation and transcriptional activation at single-cell resolution. Another future challenge will be to assess how differences in magnitude and frequency of fluctuations in otherwise phenotypically identical cells could potentially affect cellular fate. Fluctuations in gene expression are known to alter cell-fate decisions⁶⁴, and YAP has been implicated in stem cell pluripotency.⁶⁵ Transient reversals in YAP localization acting as a fast trigger of YAP activity could be relevant for spontaneous cell fate switching in stem cells as well as lineage reprogramming.^{66,67} Combining real-time detection of dynamic fluctuations in YAP localization with single-cell genomics^{68,69} and lineage tracing^{70,71} will help identify long term effects of YAP fluctuation on cell fate decisions.”

Minor Comments

Minor comment 1. As mentioned above, the authors use activated Ras or a Src inhibitor (PP1) to either decrease or increase LATS activity respectively. If the goal is to understand the contribution of LATS, then these are not good ways to do it since both treatments effect many things besides LATS. Why not knock down LATS to look at the effect of reduction in LATS activity?

Response. Per the reviewer’s suggestion, we have investigated SOCE induced YAP localization-reset in cells lacking LATS kinases. We obtained Lats1/2 knockout MCF10A cell line from the Gumbiner lab and carried out immunofluorescence based analysis of YAP localization at different time points post-Thapsigargin treatment. These experiments revealed that LATS knockout in MCF10A effectively eliminated calcium-induced YAP localization-reset (**Figure 4e-f**). We also confirmed using immunostaining that wild type MCF10A cells showed a similar timeline of calcium-induced YAP localization reset as MCF10AYAP-GFP-KI line (**Figure 4e-f**).

Minor comment 2. I have a number of concerns regarding the tagging of endogenous YAP with GFP. For all experiments YAP is tagged at its C-terminus with GFP. This disrupts the PDZ domain binding motif at the C-terminus of YAP. This motif has been shown to be important for nuclear targeting of YAP. The authors show no data testing whether the tag is affecting the function or regulation of YAP. For example, is there any effect on YAP target gene expression? Also it is not clear if both alleles of YAP are tagged or if only one, which would make a difference for testing whether YAP function is perturbed.

Response. We share the reviewer’s concern for endogenous tagging. We want to draw the attention of the reviewer to the distinction between assaying YAP localization and YAP dependent transcription. The YAP-eGFP knockin lines were used purely to look at localization dynamics. For YAP, it is challenging to preemptively decide on which termini is ideal for tagging since the N terminus has the TEAD binding domain, which arguably is the most essential element in the modular organization of YAP. Also, several examples exist in literature where researchers have used exogenous over expressions systems where they have used either N terminal fluorophore fusion (Ege et al. 2018 Cell Systems) or C terminal fluorophore fusion (Manning et al. 2018. Current Biology) to look at YAP localization. We have now done a host of experiments to investigate YAP localization in the MCF10A^{YAP-GFP-KI} line when exposed to well-established signaling cues. These experiments clearly show that the MCF10A^{YAP-GFP-KI} line recapitulates several well-established behaviors of YAP in response to changes in actomyosin contractility (**Latrunculin B treatment, Figure 1g-h**), changes in substrate stiffness (**Figure 1e-f**), changes in density (**Figure 2c-e**), and GPCR signaling (**serum starvation and replenishment or LPA treatment, Figure 3e-f**). Despite this, there still may be subtle effects on the ability of YAP to interact with its binding partners that can only be tested in a context specific manner. We have now explicitly stated this in the manuscript.

All transcription reporter experiments were done in MCF10A cells with native YAP. Therefore, our measurements of transcription activity are not impacted by any potential perturbation due to the tagging of YAP. Further, we validated the response of YAP-target genes to calcium-induced YAP localization-reset using RT qPCR measurements (**Figure 6m**) as well as single-molecule RNA FISH (**Figure 6g-l**).

Minor comment 3. The authors state “Simultaneous treatment of Ionomycin and the potent phosphatase inhibitor Okadaic acid, which inhibits protein phosphatase 1 upstream of YAP³⁹, led to a similar degree of cytoplasmic sequestration but with minimal recovery, suggesting that Ca²⁺ drives phosphorylation of YAP followed by PP1-dependent dephosphorylation (Supplementary Figure 2a).” This statement is a large over interpretation of their data. There is no data presented regarding Ca²⁺ having an effect on YAP phosphorylation under their conditions, and okadaic acid inhibits various phosphatases besides PP1 and could be acting in many different ways. Their model could be stated as one possible interpretation of the data.

Response. We thank the reviewer for pointing this out. We have now reworded the aforementioned sentence. Here’s the relevant excerpt from the main text.

“This result hints at the possibility that Ca²⁺ may drive the phosphorylation of YAP, followed by PP1-dependent dephosphorylation.”

Further, we have now shown that calcium driven YAP localization-reset is entirely abolished in MCF10A cells where LATS1/2 have been knocked out (**Figure 4e-f**). This new data provides direct evidence of a strict dependence of YAP localization-reset on LATS kinases.

Minor comment 4. The authors state that the Thapsigargin treatment mimics the effects observed in the wound healing assay. This is not quite true. In the wound healing assay, YAP quickly localizes to the nucleus then leaves, followed by slow accumulation in the nucleus. The Thapsigargin treatment causes rapid exit of YAP from the nucleus followed by slow accumulation. The authors should make clear that calcium release just mimics part of the wound healing response. Do the authors have any thoughts about what triggers the initial rapid localization of YAP to the nucleus following wounding?

Response. We thank the reviewer for pointing this out. We have now clearly stated in the text that fast calcium wave is one major components of the wound response, which Thapsigargin treatment mimics. The other critical signaling inputs are a large mechanical deformation/ stretching of cells at the wound edge and an immediate change in local cell density. Both of these inputs are major modulators of YAP localization. The YAP dynamics at the wound edge is, therefore, most likely a cumulative response to at least these three signaling inputs. We have now clarified this in the main text.

Minor comment 5. Explanation is sometimes lacking, making it difficult to tell how something was done and what the data is showing. Several examples of this are shown below.

Response. We recognize that the original manuscript was written in a condensed manner. We have clarified many aspects of the manuscript for ease of interpretation and provide a better flow for readers.

Minor comment 6. The authors state “Immunofluorescence imaging in glioblastoma cells by Liu et al showed that LATS1/2 is critical to Ca²⁺-driven phosphorylation and cytoplasmic sequestration of YAP. Indeed, treatment with the potent protein kinase C inhibitor Go697641 delayed depletion and reduced the magnitude of the localization response (Supplementary Figure 2c, Supplementary Movie 6).” The logic connecting these two sentences is not clear. What the authors are thinking should be spelled out in more detail.

Response. We thank the reviewer for pointing this out. We have now elaborated the basis for this line of experimentation more clearly. Here’s the relevant excerpt from the main text.

“A recent paper demonstrated that Protein kinase C (PKC) beta II mediates Ca²⁺-driven phosphorylation and cytoplasmic sequestration of YAP⁴⁸ in glioblastoma cells, through activation of Lats 1/2 kinases. While the role of PKCs in YAP regulation has been previously explored, the nature of the regulation seems to be context and cell type specific.⁴⁹ For example, TPA induced activation of PKCs have been shown to dephosphorylate YAP causing nuclear accumulation in Hek293a, HeLa and U251MG cells, but have the exact opposite effect on Swiss3T3 cells, MEF cells and the lung cancer A549 cells.⁴⁹ To test the role of the PKC-LATS axis in mediating calcium-driven localization reset, we treated MCF10A^{YAP-GFP-KI} cells with the potent protein kinase C inhibitor Go6976.^{48,49} This caused a distinct delay in the onset of YAP nuclear depletion and reduction in the magnitude of the localization-reset response (Supplementary Figure 3c, Supplementary Movie 8).”

Minor comment 7. What does “neighbors” mean in figure 1D and 1F? This presumably reflects cell density, but it should be spelled out in the legend what it means.

Response. Thank you. We have clarified this.

Minor comment 8. Figure 2D measures the “Fraction of cells with at least one fluctuation”. The legend says that this measures fluctuations per unit area, but over what time period?

Response. Thank you. We have clarified this.

Reviewer #3 (Remarks to the Author)

In this study Franklin et al set out to examine the dynamics and subcellular localization of YAP transcription factor in living cells in conjunction with the transcriptional activity of its target genes, to understand the circuit of activity of YAP dependent gene regulation. The system is based on endogenously tagged YAP proteins as well as the mRNAs transcribed from two chosen target genes (using CRISPR tag knock-ins). This study finds that the nuclear localization of YAP is cell density dependent and that YAP fluctuates between the nucleus and the cytoplasm. These fluctuations can be modulated by interfering with the signal transduction pathway, by administering a wound to the monolayer, and by intracellular calcium release. Intriguingly, the authors noticed fluctuations in nuclear size correlating with the changes in calcium concentration and went on to test whether changing the rigidity of the nucleus with a lamin mutant would influence YAP fluctuations. This was indeed the case.

The authors then followed transcriptional activity of the two target genes using MS2 labeling of the transcripts and infer from the YAP localization data to the transcriptional activity in interphase and after mitosis. It is quite a pity that YAP localization and transcription are not examined in the same cells. Last part of the paper was hard to follow and seems like this was not “milked” to the end. Altogether, the data are of very high quality, novel both technically and scientifically. The experiments performed cover much ground and give a broad picture of the activity and regulation of this signaling pathway in real time. Basically, this study has the potential for Nature Communications but there are some issues that I have with the manuscript in its current condition:

We thank the reviewer for appreciating the quality, and technical and scientific novelty of our work and providing lucid suggestions regarding how to improve the manuscript.

Comment 1. The authors do not do a good job in explaining or interpreting the data and really one has to spend too much effort trying to understand what is it they actually mean to say. They throw a huge amount of data on the page (described succinctly) but don't make much of an effort to go deeper and tell us what it is we learn from this study (seem to skim over the findings), or what are the mechanistics behind the phenotypes.

Response. We thank the reviewer for pointing this out. We now provide considerably more detailed explanations of experimental design, findings, and conclusions.

Comment 2. Part of the problem is that the paper is prepared Nature style and this really does not do justice to the study. I would recommend focusing on elaborating, better explanations of the data and figure legends. (In my opinion, less use of elaborate vocabulary and unusual terminology in the Abstract and Introduction would also help the situation). The paper now has 4 main figures. Probably this could be broken down now, panels enlarged, additional supporting information added, all in all trying to make the study more intelligible to the reader.

Response. We thank the reviewer for pointing this out. The manuscript is now ~5300 words long, has **8** main text figures (**32 new main text Figure panels**), **2** new Supplementary Figures (**7 new supplementary figure panels**) and **5** new Supplementary Movies, which represents a cumulation of a host of new experiments and also a strategic redistribution of earlier figure components to help ease the flow of the paper.

Comment 3. Notably, there is no supplementary molecular or biochemical data (genomic PCR, RT-PCR, Western etc.) to persuade us that the knock-in's worked as expected in all the cell lines, that HRAS is overexpressed in the cells etc. We are expected to take all this for granted but that's not the way things are done. For example, the insertion of both GFP and puromycin – does this affect the final protein (is the scheme drawn to scale)? What about some control experiments with endogenous YAP using antibodies to show that one sees similar phenotypes of the endogenous and fusion proteins, or RNA FISH for the transcription part?

Response. We thank the reviewer for pointing out the omission of control experiments in the earlier submission that we had done to validate knockin efficiency. We have now included genomic PCR validation of site-specific knock-in and a western blot, which clearly shows that the fusion protein has the expected molecular weight of YAP-eGFP. Since puromycin is separated from GFP by a P2A peptide, we did not expect puromycin to interfere with YAP activity. Further, we did not detect any larger band in the western blot representative of YAP-eGFP-puromycin fusion, indicating puromycin is efficiently cleaved by the self-cleaving P2A (porcine teschovirus-1 2A) peptide.

Our MCF10AT cell line, where HRasG12V is overexpressed, has been validated extensively in previous work (Hu et al. 2008, Cancer Cell; Imbalzano et al. 2009, Cancer Cell Int.; Levental et al. 2010. Cell; Shi et al. PNAS 2016).

As suggested by the reviewer, we performed further validation of YAP localization using immunostaining of native YAP in MCF10A-^{YAP-eGFP KI} lines. These experiments showed a robust correlation of the YAP nucleocytoplasmic ratio between YAP-eGFP and antibody staining (Figure 1c-d). As suggested by other reviewers, we did further perturbations and control measurements (see above), which all align with previous reports.

Single-molecule RNA fish against native genes can be finicky, especially if the levels of gene expression are low, or probes have not been optimized for particular genes of interest. Therefore, to achieve a population-level perspective of the impact of YAP localization reset on transcription, we performed RT qPCR analysis of transcription of several established YAP target genes (AREG, Ankrd1, CTGF, IGFBP3, TGFB2, Foxf2, RASSF2, CYR61) and housekeeping genes (GAPDH, HPRT1, PGK1) in a time-course after intracellular calcium release by thapsigargin. These measurements confirmed that the genes probed in our nascent transcription cell lines (AREG and ANKRD1) both increased at the population level following calcium release. Our results also showed that the effect was consistent with several other YAP target genes (**Figure 6m**) but did not change levels of housekeeping gene expression in an appreciable way (**Figure 6m**).

We further validated the MS2 knockin specificity by dual-color analysis of nascent transcription spots visualized using MCP-mNeon in AREG-MS2 line and single-molecule RNA FISH against native AREG transcription (**Figure 5b**).

Comparison of changes in nascent transcription upon induction of YAP localization reset using single-molecule RNA FISH against native AREG and GAPDH further confirmed that the effect was specific to AREG (**Figure 6g-l**).

Other points:

Point 1. Abstract – “concerted localization resets” – the reader has no way of understanding this peculiar phrase when reading the abstract. In fact the term “localization reset” is never explained. The whole sentence is even more confusing –“ concerted localization resets, which are dramatic, concerted departure/reentry cycles”

Response. We agree with the reviewer about the complexity of the phrase, especially as a stand alone phrase in the abstract. We have now addressed this in the manuscript.

Point 2. Fig. 1C – unclear what is “counts” in the y-axis and it takes time realize that the densities on the right plots refer also to the left hand plots (legend does not give much explanation).

Response. We thank the reviewer for pointing this out. This plot in Figure 2 is now changed to ‘cell counts’, and the cell density is written on each panel. At each density, the same images were quantified for YAP and TEAD N/C. We have explained this in the legend.

Point 3. Fig. 1D – “neighbors” in axis - supposedly you mean no. of neighboring cells? Is it possible that a cells has 10’s of neighbors?

Response. We define neighbors as the number of cell nuclei centroids within a radius of ~40 um. This was defined in the legend. We have changed the label to “local cell density” to better describe the metric.

Point 4. Movie 1- the density in the second part of the movie does not reach the same levels as in the first part. As for cell density – there could have been more analysis of this issue.

Response. We have found that RAS transformed cells differ in growth rate and do not grow to the same high packing density as wild type MCF10A. We, therefore, plotted our time-lapse data for monolayer growth over the *same range of local cell densities* to account for the difference in growth rates, morphologies, and packing densities (**current Figure 2d-e**). This shows us that for equivalently **high** cell densities, we observe strong YAP cytoplasmic sequestration in MCF10A and N/C~1 for MCF10A + HRas. We have now explained this reasoning in the manuscript:

“Interestingly, we found that HRas transformation alters the packing density of MCF10A, in addition to its known effect on growth rate (Supplementary Movie 2). To account for these changes, we plotted the YAP N/C measurements over the same range of local cell densities for MCF10A versus HRas (Figure 2c and 2e). This clearly shows that at equivalent high density, HRas transformed cells have impaired nuclear exclusion.”

Point 5. Fig. 2B – no numbers on y axis

Response. Current figure 3b. We have now added the N/C scale on the y-axis.

Point 6. Fig. 2C – no explanation. What do the plots tell us? How can there be a 0 for N/C?

Response. In the caption, we explained that this data has the mean N/C value subtracted from individual tracks. This was important because it allows normalizing the data across the sample for visual comparison.

Point 7. Fig. 2E – how frequent is this phenomenon?

Response. In our initial submission we had stated that this a rare phenomenon. We did not observe this frequently enough to perform statistical analysis. Since this observation is rare, and does not add in any significant way to the central tenet of this work, we have deleted this figure and corresponding textual reference in the current submission.

Spontaneous coordinated fluctuation is however an interesting collective dynamic phenomenon that would be of particular interest to investigate in the context of multicellular structures in future works.

Point 8. Fig. 2G – only one row of cells was influenced? If so, this should be mentioned.

Response. We originally wrote in the main text that the edge cells were fluctuating. The edge cells are indeed the single row of cells.

Here's the relevant excerpt from the main text:

“Surprisingly, real-time tracking of YAP immediately after wounding the monolayer revealed an oscillatory response: rapid nuclear accumulation in edge cells (~1 minute), followed by depletion (lasting ~20 minutes), and finally slow nuclear re-accumulation (~3 hours, Figure 4a, Supplementary Movie 7).”

Point 9. Fig. 2H – are these frames from a time lapse movie? Are the times in 2J the same for all? Is the first image before or after treatment? Line 118 – “resulting in a bulk YAP translocation cycle similar to that seen at the wound edge” – actually it's not the same. Importantly, it is a pity that the calcium and YAP are not imaged together in the same cell (and to show this also for the wounding experiment).

Response.

2H: Yes. These are individual frames from a time-lapse movie (current **Figure 4b**).

2J: Yes. The times written into the images are the same for 2H, 2I, 2J (current **Figure 4b, 4c, 4d**).

last submission Line 118: We thank the reviewer for pointing this out. We have now added more detail to the text about the differences.

Here's the relevant excerpt from the manuscript explaining this:

“The rapid fluctuations in YAP localization upon wounding hinted at a fast-acting upstream signaling pathway. Both mechanical deformation of cells at the wound edge and a dramatic shift in local cell density could potentially contribute to the oscillation in YAP localization. A fast calcium wave (FCW) within seconds of epithelial wounding has been reported in diverse cell types, including MCF10A.^{40,41,42} We hypothesized that intracellular calcium may play a role in the rapid nuclear accumulation of YAP upon wounding. To test this possibility, we released intracellular calcium via Thapsigargin (TG)⁴³ treatment. This store operated calcium entry (SOCE) resulted in a bulk YAP translocation cycle: an initial fast depletion (lasting ~25 minutes) followed by slow nuclear enrichment (50 minutes, Figure 4b, Supplementary Movie 8). Although the wound assay showed a distinct pattern with an initial phase of nuclear translocation, TG treatment helped us isolate the effects of transient Ca²⁺ release on YAP from the multitude of parallel signals occurring during the wound response.”

Simultaneous Calcium and YAP imaging: The most ideal reporter of fast kinetic changes in Calcium level is GcAMP6f, which we have used in this paper. Unfortunately, GcAMP6f has the same excitation/emission spectra as eGFP, precluding simultaneous dual-color imaging.

However, since the response is so uniform for calcium and YAP across cells, we did not feel that dual-color simultaneous imaging of YAP and calcium was warranted.

Point 10. Line 97 – “The fluctuation magnitude and frequency were unchanged across conditions” – this remains as a statement with no elaboration or conclusion and these parameters are not introduced in any way. Actually, PP1 shows some difference.

Response. We have updated the text to more thoroughly explain the parameters and differences.

Here's the relevant excerpt from the main text.

“Further, we characterized the frequency (number of detected fluctuations per time) and amplitude (net change in N/C) of these fluctuations. The fluctuation frequency and amplitude were unchanged upon HRas transformation (Supplementary Figure 2a-b). On the other hand, PP1 treatment led to a slight, but significant, increase in both fluctuation frequency and amplitude (Supplementary Figure 2a-b).”

Point 11. Line 99 – PP1 – no explanation what this is

Response. We have now explained the function and utility of inhibiting Src kinase with PP1.

Point 12. Movie 3 – don't really see any reduction in N/C in control cells. Maybe add arrows to point to the cells since its moving so fast.

Response. We have updated movie three. For control cells, we have animated the nuclear outlines on a particular cell that shows a dramatic YAP fluctuation.

Point 13. Line 103 – “fast localization fluctuations of YAP” – how fast?

Response. We have changed the text in for this section. Now we provide specific time scale of the fast localization changes.

Here's the relevant excerpt from the main text.

“Single cell localization traces revealed rapid changes in YAP N/C ratio (time scale of 30-60 minutes), suggesting that YAP localization is dynamically tunable (Figure 3b, c).”

Point 14. Fig. 3 – Here too no controls. For instance, a smFISH experiment to show how the gene acts under regular conditions prior to knock-ins and that it doesn't change after the knock-in. Importantly, as mentioned above, show the RNA and protein in the same cell. If this is hard to do in live cells, then at least in fixed cells it can be shown (Ab to YAP in gene-MS2 cells, and smFISH in the YAP-eGFP cells). What about a control showing an unaffected gene under same conditions and treatments (doesn't have to be MS2 tagged).

Response. Per suggestion by the reviewer, we have added additional critical controls for our transcription measurements.

Figure 5b: Co-localization of AREG-MS2 nascent spots visualized using MCP-mNeon and simultaneously labeled with Quasar® 670-labeled smRNA FISH probes against the native AREG transcript.

Figure 6g-l: Images and quantification of nascent transcription using Quasar® 670-labeled smRNA FISH probes against AREG and housekeeping gene GAPDH in wild type MCF10A cells upon Thapsigargin treatment. This was critical for showing that the calcium-induced gene expression occurs in unedited cells, and is specific to YAP target AREG, as it did not significantly alter GAPDH expression. We extended this analysis further by carrying out RT qPCR analysis of transcription of 8 different YAP target genes and 3 different housekeeping genes during a Thapsigargin treatment time course of native unedited MCF10A cells.

To show protein dynamics and transcription dynamics in the same cell we would need to generate a triple stable cell line expressing MCP-FP under inducible control and two different CRISPR knockins. This is inherently difficult with MCF10A cells, where we are limited in the number of genetic alterations and selection rounds, as this cell line begins to exhibit morphological changes. With our AREG and Ankrd1 transcription reporter cell lines we have noticed phenotypic transformations beyond 10 passages of culture.

While simultaneous visualization of protein dynamics and transcription kinetics in the same cell is a very relevant next step, it is beyond the scope and timeline of the current work, as we would need to establish the dual knockin in a cell line that is more resistant to passage based transformation and validate its functionality. We have added this as an important future direction in the discussion section.

Point 15. Supplementary Figure 4 – no real explanations or interpretation/discussion

Response. We have more clearly explained the results of Supplementary Figure 4 (**Current Supplementary Figure 5**) in the main text. Here's the new excerpt where we refer to this Supplementary Figure 5 multiple times in much more detail.

“Interestingly, we found that TG treatment rapidly increased both the number of cells transcribing ANKRD1 and AREG (Figure 6a-b; Supplementary Movie 12) and the intensity of nascent spots (Supplementary Figure 5a). The calcium induced localization-reset of YAP and concomitant up-regulation of YAP dependent transcription, point towards a non-linear relationship between YAP localization and target gene activation. This is in contrast to the standard model equating nuclear enrichment with YAP activity. To test the potential connection between localization-resets and transcription, we blunted the TG-induced localization-resets with Go6976 pre-treatment, which severely attenuated both AREG and ANKRD1 transcriptional responses in terms of fraction of transcribing cells (Figure 6c-d). Further, we found no significant increase in transcription pulse intensity or duration (Supplementary Figure 5a-b). To further test a connection between YAP localization-resets and native transcriptional responses, we hypothesized that the Src inhibitor PP1 would also increase YAP responsive transcriptional outputs, based on PP1’s ability to increase YAP N/C fluctuations. Indeed, both ANKRD1 and AREG showed moderate increase in transcription following 1μM PP1 treatment (Figure 6e-f). PP1 treatment led to an immediate (albeit moderate) increase in ANKRD1 while AREG increased more slowly peaking ~7 hours post treatment (Figure 6e-f). Additionally, PP1 treatment increased the transcription pulse intensity of ANKRD1 (Supplementary Figure 5a) and the number of long-lived transcription pulses for AREG (Supplementary Figure 5b).”

Point 16. Figure 4 –nothing here is well explained. Lots of data condensed together – really one cannot easily follow or try to review this part (not to mention the non-regular presentation of the FRAP data that needs explaining) and understand what is the biological significance of this information.

Response. We have expanded our explanation of design and rationale of photobleaching experiments for measuring YAP transport rates as well as spatiotemporal FRAP for measuring YAP chromatin interaction.

Here are two excerpts providing more detail behind the experimental rationale.

“To measure first-order nuclear import and export rate of YAP, we carried out fluorescence recovery after photobleaching experiments, where we monitored eGFP fluorescence in the nucleus after bleaching either the nuclear or cytoplasmic signal, in cells expressing native YAP-eGFP or an inert nuclear transport cargo (NLS-eGFP_{2x}-NES) (Figure 8b).”

“To understand whether malignant transformation affects YAP-chromatin interactions in breast epithelial cells, we performed high time-resolution spatiotemporal FRAP experiments on native YAP-eGFP in the nucleus. Briefly, a 2 pixel wide line bisecting a circular bleach spot was scanned repeatedly at high speed (~220 Hz) to generate radial intensity profiles of fluorescence recovery at a high temporal frequency. This allowed for more accurate fitting than in case of conventional FRAP, which is critical for binding and diffusivity analysis of fast exchanging TFs^{61,62,63}.”

We have also explained the results of the spatiotemporal FRAP in more detail.

“We measured an effective binding equilibrium close to 1, such that at any time point, approximately 50% of YAP molecules were effectively bound. Because the equilibrium constant is close to 1, this bulk interaction mode is relatively weak, but it is globally present throughout the nucleus.

Consistent with the extremely rapid FRAP recovery of YAP in the nucleus (Figure 8d), all three oncogenically transformed lines were best described by a pure diffusion model (Supplementary Figure 7, Supplementary Table 1). These results suggest that bulk interaction in the nucleus, most likely to chromatin, could restrain YAP nucleocytoplasmic transport in non-transformed cells through an unidentified binding partner. Whereas transformation abolishes this interaction, which could either be specific to Ras activation or a more general oncogenic signaling state.”

Point 17. Discussion – “chromatin interactions” - not really something measured in this study

Response. We indeed measured this through spatiotemporal FRAP modeling. This is commonly used to assess interactions between a transcription factor/co-activator and chromatin. There has been a paper published on YAP-chromatin interactions measured by FRAP, as referenced in the main text (Ege et al. 2018). We have more clearly explained the rational and significance of these measurements. See response to Point 16.

Point 18. Number the movie files.

Response. We have numbered the movie files.

REVIEWERS' COMMENTS

Reviewer #1 (Remarks to the Author):

The authors performed many of the requested experiments, which reinforced the usefulness of their new system, and their claims on the novel observation about the "resets". The connection with calcium remains somewhat of an interesting but poorly developed story, and I would strongly recommend to add in the discussion some cautionary statement on the current lack of evidence that physiologically-relevant calcium variations do affect YAP/TAZ localization. Overall the manuscript is now suitable for publication.

Reviewer #2 (Remarks to the Author):

The authors have addressed my primary concerns and the revised manuscript is much improved. I have a couple small details that should be addressed prior to publication.

1) It is not clearly spelled out whether both alleles of YAP (and TEAD) are tagged with GFP. Supplementary figure 1B seems to indicate that the cells have both tagged and untagged alleles. This should be stated explicitly in the text.

2) The authors state: "We further observed a steady accumulation of membrane-localized ANKRD1 transcripts following serum stimulation (Figure 5h-i, Supplementary Movie 11)." It is not clear how the images or movie shows this. Perhaps the "membrane-localized" transcripts could be indicated in the figure/movie.

Reviewer #3 (Remarks to the Author):

The authors did a good job at answering the reviewers many queries and now the paper is better structured, and in my opinion easier to follow. I think the study is important and broad, and I recommend publication.

A few things that caught my eye while reading:

Abstract line 24 - 'transformed' appears twice

Fig. 1e - what is Sir-DNA?

Fig. 3h - would it be possible to add an average?

Fig. 5h - I suggest clarifying what is the accumulating signal that is increasing after serum addition. Hard to detect what that is in the image. Also, is it possible to quantify what is the time frame between the addition of serum and the appearance of the RNAs in the cytoplasm?

Discussion - it is a pity this section is somewhat short, much more could have been discussed, also in the context of other studies with similar directions. For instance, the changes in YAP localization during serum starvation/release can be compared to other signaling pathways in which proteins enter the nucleus and change the transcriptional activity of genes, for example there are studies on p53 fluctuations, or the effect of serum on the activity of the beta-actin gene. Another point could be the pattern of transcription after cell division - there is a paper by David Spector's group looking in a very similar manner at transcription in living cells right after mitosis.

Reviewer's comments (Revision stage 2)

Reviewer #1 comment

The authors performed many of the requested experiments, which reinforced the usefulness of their new system, and their claims on the novel observation about the "resets". The connection with calcium remains somewhat of an interesting but poorly developed story, and I would strongly recommend to add in the discussion some cautionary statement on the current lack of evidence that physiologically-relevant calcium variations do affect YAP/TAZ localization. Overall the manuscript is now suitable for publication.

Response

We thank the reviewer for endorsing of our work.

We have now added a disclaimer in the discussion section about the need for further work in establishing the physiological relevance of variations in calcium level on YAP/TAZ localization.

Here is the relevant section from the text:

"While our data clearly show that SOCE induces YAP localization reset through the LATS kinase regulatory axis, it is imperative to investigate the effect of transient physiological changes in Calcium levels on YAP localization in various cellular contexts, especially how such changes may affect YAP target genes in the long term."

Reviewer #2 comment

General Comment

The authors have addressed my primary concerns and the revised manuscript is much improved.

Response

We thank the reviewer for endorsing of our work.

I have a couple small details that should be addressed prior to publication.

Comment 1. It is not clearly spelled out whether both alleles of YAP (and TEAD) are tagged with GFP. Supplementary figure 1B seems to indicate that the cells have both tagged and untagged alleles. This should be stated explicitly in the text.

Response

We thank the reviewer for pointing this out. We have now clarified this in the main text. Here's the relevant section:

"Genomic PCR showed proper insertion of an eGFP-p2a-puromycin cassette at the C-terminus of YAP (Supplementary Figure 1b, Methods, Supplementary Table 2), and revealed amplicons of two different sizes corresponding to knockin and wildtype alleles suggesting that the base MCF10A^{YAP-GFP-KI} cell line is a mixture of heterozygous and homozygous knockins."

Comment 2. The authors state: "We further observed a steady accumulation of membrane-localized ANKRD1 transcripts following serum stimulation (Figure 5h-i, Supplementary Movie 11)." It is not clear how the images or movie shows this. Perhaps the "membrane-localized" transcripts could be indicated in the figure/movie.

Response

We have added arrowheads showing particle accumulation in Figures 5h.

Reviewer #3 comment

General comment

The authors did a good job at answering the reviewers many queries and now the paper is better structured, and in my opinion easier to follow. I think the study is important and broad, and I recommend publication.

Response

We thank the reviewer for endorsing of our work.

Specific comments

Comment 1.

Abstract line 24 – 'transformed' appears twice

Response

We have corrected this.

Comment 2.

Fig. 1e – what is Sir-DNA.

Response

We have provided details about SiR-DNA in the figure legend and in the methods section.

Comment 3.

Fig. 3h – would it be possible to add an average?

Response

The intention of this measurement is to highlight the rapid fluctuations of YAP localization in human embryonic stem cells. Since fluctuations are stochastic and rarely coordinated between cells, an average value would not be meaningful in terms of any physiologically relevant phenomena.

Comment 4.

Fig. 5h – I suggest clarifying what is the accumulating signal that is increasing after serum addition. Hard to detect what that is in the image. Also, is it possible to quantify what is the time frame between the addition of serum and the appearance of the RNAs in the cytoplasm?

Response

We have added arrowheads showing particle accumulation in Figures 5h. We now quantify the increase in cytoplasmic transcript accumulation over the course of the three hours post serum addition. This shows a statistically significant increase in transcripts at about 1.5-2hrs post serum, with a further increase transcript counts after 2.25-2.9hrs. We thank the reviewer for asking about this. We have now further elaborated on the dynamics of the response in Figure 5i and the main text.

Comment 5

Discussion – it is a pity this section is somewhat short, much more could have been discussed, also in the context of other studies with similar directions. For instance, the changes in YAP localization during serum starvation/release can be compared to other signaling pathways in which proteins enter the nucleus and change the transcriptional activity of genes, for example there are studies on p53 fluctuations, or the effect of serum on the activity of the beta-actin gene.

Response

We thank the reviewer for the suggestion. We have added new discussion material along the line of the suggestion.

Here's the relevant section from the manuscript.

“Transcriptional pulses are regulated combinatorially, where the response is dependent on the appropriate levels and localization of multiple effectors. For example, in the case of the β actin gene, the appropriate transcriptional response to serum-induced signaling is dependent on the right balance of serum response factor and actin⁶⁴. MAL, a monomeric actin-binding protein, that shuttles between nucleus and cytoplasm, accumulates rapidly in the nucleus upon serum stimulation⁶⁵, similar to the rapid shift in YAP localization upon serum starvation and replenishment. Identifying molecular partners that can fine-tune YAP localization resets in a context-specific manner is an important future direction. While oscillations in localization of many transcription factors such as NFAT, NFkB, p53, Wnt, have been connected to transcriptional response dynamics in a wide variety of cellular contexts^{11,12,13,66,67,68}, one future challenge will be to assess how differences in magnitude and frequency of fluctuations in otherwise phenotypically identical cells could potentially affect cellular fate.”

Comment 6

Another point could be the pattern of transcription after cell division - there is a paper by David Spector's group looking in a very similar manner at transcription in living cells right after mitosis.

Response

We thank the reviewer for the suggestion. We have added new discussion material along the line of the suggestion.

Here's the relevant section from the manuscript.

“Our data show a transient but robust spike in the transcription of select YAP target genes upon mitotic exit, suggesting a possible role of YAP in mitotic bookmarking, which is a fast reboot mechanism for the expression of genes involved in metabolism and cell survival following cytokinesis^{77,78,79}. While YAP itself shows marked exclusion from condensed chromatin during mitosis, the effective YAP localization reset during early stages of mitotic exit, may initiate new rounds of engagement between transcriptionally competent YAP and chromatin localized TEAD. Whether YAP is a dynamic bookmarking factor that promotes postmitotic expression of YAP target genes across the board, needs to be explored, especially in the context of stem cell renewal and embryonic development.”